# The Hedgehog & the Porcupine: Expressive Linear Attentions with Softmax Mimicry

**Michael Zhang, Kush Bhatia, Hermann Kumbong, and Christopher Ré**
Department of Computer Science, Stanford University
{mzhang,kushb,chrismre}@cs.stanford.edu, kumboh@stanford.edu,

## Abstract

Linear attentions have shown potential for improving Transformer efficiency, reducing attention's quadratic complexity to linear in sequence length. This holds exciting promise for (1) training linear Transformers from scratch, (2) "finetuned-conversion" of task-specific Transformers into linear versions that recover task performance, and (3) "pretrained-conversion" of Transformers such as large language models into linear versions finetunable on downstream tasks. However, linear attentions often underperform standard softmax attention in quality. To close this performance gap, we find prior linear attentions lack key properties of softmax attention tied to good performance: low-entropy (or "spiky") weights and dot-product monotonicity. We further observe surprisingly simple feature maps that retain these properties and match softmax performance, but are inefficient to compute in linear attention. We thus propose Hedgehog, a learnable linear attention that retains the spiky and monotonic properties of softmax attention while maintaining linear complexity. Hedgehog uses simple trainable MLPs to produce attention weights mimicking softmax attention. Experiments show Hedgehog recovers over 99% of standard Transformer quality in train-from-scratch and finetuned-conversion settings, outperforming prior linear attentions up to 6 perplexity points on WikiText-103 with causal GPTs, and up to 8.7 GLUE score points on finetuned bidirectional BERTs. Hedgehog also enables pretrained-conversion. Converting a pretrained GPT-2 into a linear attention variant achieves state-of-the-art 16.7 perplexity on WikiText-103 for 125M subquadratic decoder models. We finally turn a pretrained Llama-2 7B into a viable linear attention Llama. With low-rank adaptation, Hedgehog-Llama2 7B achieves 28.1 higher ROUGE-1 points over the base standard attention model, where prior linear attentions lead to 16.5 point drops.

## 1 Introduction

Linear attentions are promising methods for improving Transformer efficiency. By replacing the softmax of attention's query and key dot products with kernel function feature maps, linear attentions reduce attention's time and space complexity from $\mathcal{O}(n^2d)$ to $\mathcal{O}(ndd')$ where $n$ is sequence length, $d$ is head dimension and $d'$ the feature map dimension (Katharopoulos et al., 2020; Choromanski et al., 2020; Peng et al., 2021; Xiong et al., 2021; Schlag et al., 2021). For typical Transformer settings, *e.g.,* with head dimension = 64 and sequence lengths at 512 to 32K, this quadratic-to-linear scaling can result in significant speed and memory improvements (Fig. 6). As drop-in alternatives to popular softmax attention (Vaswani et al., 2017), linear attentions not only improve Transformer efficiency when training new models from scratch but can also improve inference efficiency by converting pretrained Transformers into corresponding linear variants (Kasai et al., 2021; Mao, 2022). Linear attention enables efficient Transformers in a variety of regimes:

- **Training-from-scratch**: training Transformer models with linear attention with the goal of matching standard Transformer performance, *e.g.,* as tested on benchmarks such as Long Range Arena (LRA) classification (Tay et al., 2021) and WikiText-103 language modeling (Merity et al., 2017).

- **Finetuned-conversion**: swapping the attentions of task-specific Transformers and finetuning them to convert existing models into linear versions, with the goal to recover original task performance with improved efficiency (Kasai et al., 2021; Mao, 2022).

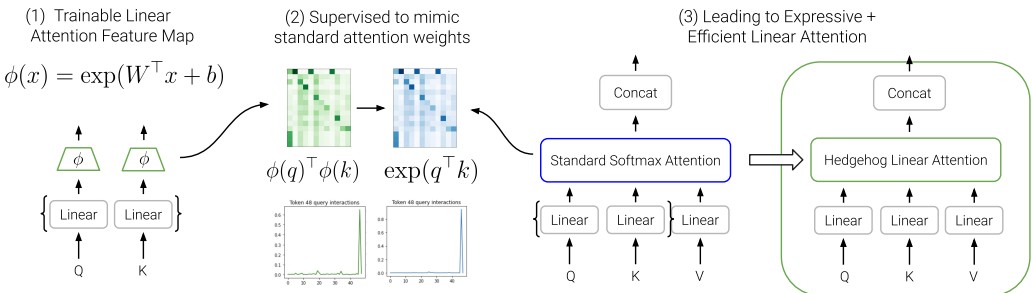

Figure 1: Hedgehog learns a trainable linear attention feature map designed to mimic standard attention, resulting in expressive yet efficient linear attentions for various Transformer training settings

- **Pretrained-conversion**: doing the same as finetuned-conversion but for pretrained Transformers such as large language models (LLMs), *e.g.,* to transfer to new tasks and longer contexts.

Unfortunately, existing linear attention mechanisms typically fail to match softmax attention in modeling quality. When training from scratch, linear attentions achieve 4-6 worse perplexity (ppl) than softmax attention on standard benchmarks such as WikiText-103 (Schlag et al., 2021; Irie et al., 2021; Fu et al., 2023), the equivalent gap between 125M and 255M Transformers (Dai et al., 2019). When converting finetuned models, linear attention models require additional quadratic attention modules to close the gap (Kasai et al., 2021; Mao, 2022). One might worry that such gaps are fundamental; for example, recent theory using the Strong Exponential Time Hypothesis (SETH) showed that high-quality truly subquadratic algorithms to approximate softmax attention may be impossible with large sequence length $n$ (Alman & Song, 2023; Keles et al., 2023).

We begin by empirically studying why this performance gap exists between standard softmax and proposed linear attentions. We identify two simple properties for softmax attention which prior linear attentions lack: 1) low-entropy "spikyness" and 2) dot-product monotonicity. We hypothesize that the quality gap in linear attentions corresponds with lacking these two properties:

- **Low-entropy "spikyness"**: Intuitively, we want attentions that attend to relevant tokens while ignoring irrelevant ones via their query-key interactions. We observe these low-entropy or "spiky" attention-weight distributions in standard Transformer attention but not prior linear attention maps—where spikes enabled via the scaled dot-product softmax are lost via other feature maps (Fig. 2)—and find this strongly corresponds to Transformer performance (Fig. 4).

- **Dot-product monotonicity**: This property requires that attention weights increase as the dot products of their corresponding queries and keys increase. Intuitively, the lack of this monotonicity can produce unstable gradients during training and finetuning, where increasing the query-key dot product can result in decreasing the attention weight the other way (and vice versa).

As a first step to recover these properties, we explore simple feature maps—such as low-degree Taylor polynomial approximations to the $\exp()$ function—that satisfy the above two properties (albeit in restricted regimes of bounded query-key dot products). In practice, we find that queries and keys are often bounded, resulting in linear attentions that recover softmax attention's spikiness, monotonicity, and subsequent performance. Unfortunately, while technically linear in sequence length, these polynomial feature maps remain inefficient to compute. They take $\mathcal{O}(nd^{p+1})$ time and space, and we find degree $p \geq 2$ necessary for performance.

We thus propose Hedgehog, an efficient-to-compute *learnable* linear attention trained to capture the spiky and monotonic softmax properties. Unlike prior works that propose a specific kernel function (Katharopoulos et al., 2020; Choromanski et al., 2020; Qin et al., 2022b) and our polynomial feature maps, we learn these feature maps as single-layer MLPs specifically *trained to match* softmax attention weights. By mapping from $\mathbb{R}^d \mapsto \mathbb{R}^d$, we maintain prior linear attentions' $\mathcal{O}(nd^2)$ complexity. However, training these mappings via softmax attention weights as cross-entropy soft-labels, we find Hedgehog can match softmax attention weights with much higher fidelity (Fig. 7), producing low-entropy and monotonic weights that match standard attention performance quality.

We validate experimentally that Hedgehog's improved expressivity translates to closing the softmax attention performance gap in the three regimes mentioned above:

- **Training-from-scratch**: we find Hedgehog matches Transformers on standard attention benchmarks such as Long Range Arena (LRA) (Tay et al., 2021) task, and closes the linear attention gap by 68.6% on WikiText-103 language modeling (improving up to 6 ppl).

- **Finetuned-conversion**: we find Hedgehog recovers >99% of original model performance on average across bidirectional encoder-only 110M BERT-base models finetuned on GLUE and causal decoder-only 125M GPT models finetuned on Wikitext-103.

- **Pretrained-conversion**: we find Hedgehog enables effective transfer to new tasks and efficient scaling to longer contexts, while frequently outperforming modern subquadratic sequence architectures by linearizing existing pretrained Transformers. A 125M Hedgehog-GPT-2 finetuned on Wikitext-103 achieves a new state-of-the-art 16.7 ppl for subquadratic models of the same size.

Finally, we demonstrate that Hedgehog can be scaled up to modern large language models; we convert pretrained Llama-2 7B into a viable linear attention Llama. With low-rank adaptation, Hedgehog-Llama2 7B achieves up to 28.1 higher ROUGE-1 points over the base standard attention model. In contrast, prior linear attentions result in models that struggle to produce coherent text (with 16.5 ROUGE-1 point drops).

## 2 PRELIMINARIES AND RELATED WORK

We provide background on attention computation, describe kernel feature based linear attentions, and finally provide details on existing linear attention mechanisms proposed in the literature.

**Attention setup**. Let $\{q_i\}_{i=1}^n$, $\{k_i\}_{i=1}^n$, $\{v_i\}_{i=1}^n$ denote the set of queries, keys, and values, with individual elements in $\mathbb{R}^d$. Let $n$ denote sequence length and $d$ denote head dimension. We compute attention outputs $y_i \in \mathbb{R}^d$ by first computing similarities between each $q_i$ and every $k_j$; for causal attention we compute these similarities for $j \leq i$. The vanilla Transformer attention computes these similarities using the softmax dot products (Vaswani et al., 2017):

$$y_i = \sum_{j=1}^i \text{sim}(q_i, k_j) v_j, \quad \text{where} \quad \text{sim}(q_i, k_j) = \frac{\exp(q_i^\top k_j / \sqrt{d})}{\sum_{m=1}^i \exp(q_i^\top k_m / \sqrt{d})} . \tag{1}$$

While very expressive, computing attention via Eq. 1 for all $\{y_i\}_{i=1}^n$ requires $\mathcal{O}(n^2 d)$ time and memory, making this inefficient for long sequences. To improve efficiency without sacrificing quality, we thus want alternative *linear attention* maps which maintain standard attention's expressivity.

**Linear attention and kernel functions**. Observe that the $\exp(\cdot)$ in Eq. 1 can be viewed as a kernel function, which Tsai et al. (2019); Katharopoulos et al. (2020) show can be replaced in general with $\mathcal{K}(x, x') = \phi(x)^\top \phi(x')$. Here $\phi : \mathbb{R}^d \mapsto \mathbb{R}^{d'}$ is a feature map applied to each vector. We can thus compute attention in *linear* time and space over the sequence length $n$, seen by rewriting Eq. 1 as:

$$y_i = \frac{\phi(q_i) \sum_{j=1}^i \left( \phi(k_j)^\top v_j \right)}{\phi(q_i) \sum_{j=1}^i \phi(k_j)} . \tag{2}$$

**Prior feature maps**. From the previous section, we observe that linear attentions are promising directions for improving Transformer efficiency at both training and inference time. Numerous prior works have proposed feature maps $\phi$ aiming to remain more efficient (where linear attention is desirable to standard attention if $d' < n$), while still being expressive and stable to train. These range from $\phi$ ensuring positive attention weights, *e.g.,* via $1 + \text{ELU}$ (Katharopoulos et al., 2020) or ReLU (Kasai et al., 2021), to softmax or Gaussian kernel approximations via randomized features (Rahimi & Recht, 2007; Choromanski et al., 2020; Peng et al., 2021; Choromanski et al., 2021; Zheng et al., 2023) or low-rank approximations (Xiong et al., 2021; Chen et al., 2021).

## 3 IMPROVING LINEAR ATTENTION VIA SPIKY AND MONOTONIC WEIGHTS

We begin by identifying two key properties of attention weights which we hypothesize are essential for good performance quality. The first, *low-entropy spikyness*, requires that the attention map is able to capture effectively capture sparse relevant tokens in a sequence. The second, *monotonicity over query-key dot products*, requires the attention map to increase with increasing dot products, and allows for smooth conversion of pretrained Transformers into linear variants.

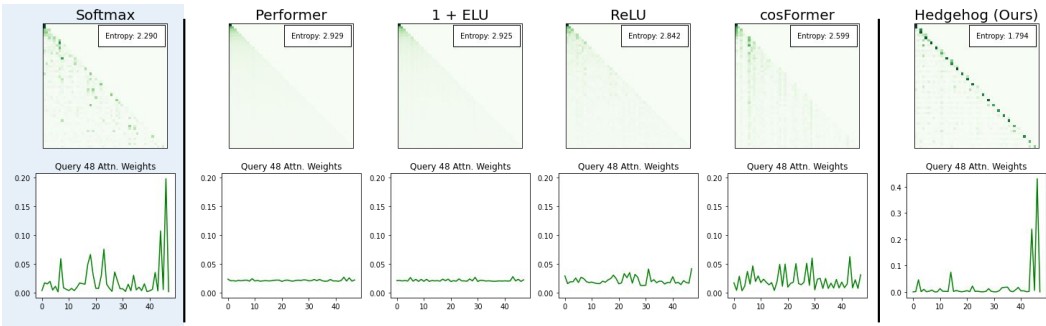

Figure 2: **Attention weight spikiness**. (Plots 1 - 5): Softmax attention results in lower entropy and "spiky" selective weighting compared to prior linear attentions (training from scratch on associative recall (Sec. 3.2)). (Plot 6): By training to mimic softmax attention, our proposed Hedgehog recovers this spikiness as a linear attention, corresponding with improved performance (Sec. 5).

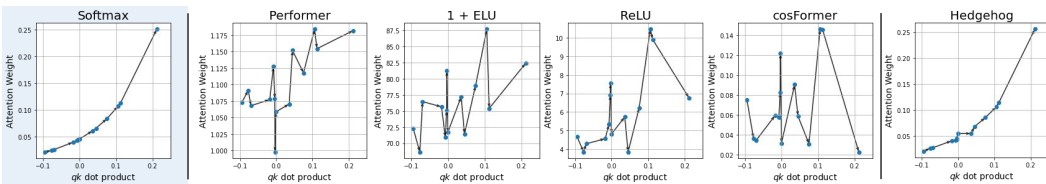

Figure 3: **Attention weight monotonicity**. (Plots 1 - 5): In contrast to softmax attention, prior linear attentions are not smoothly monotonic over trained query-key dot products, resulting in poor performance when converting BERT models by replacing attentions (Table 1). (Plot 6): Hedgehog recovers this monotonicity, and thus recovers 99% of BERT performance after conversion (Table 8).

## 3.1 PROPERTIES FOR EXPRESSIVE ATTENTION MAPS

Here we describe the spiky and monotonic properties hypothesized for desirable linear attention. We note these add to past observations for more performant linear attentions, including positive attention weights (Katharopoulos et al., 2020), orthogonal features (Choromanski et al., 2020; Irie et al., 2021), or locality (upweighting nearby values) (Qin et al., 2022a;b). We validate these properties among past linear attentions in Sec. 3.2, and preview how our proposed Hedgehog linear attention recovers these properties in correspondence with improved performance (Sec. 5) in Fig. 2, 3.

**Low-entropy spikiness.** Intuitively, one source of attention's effectiveness is its ability to selectively upweight relevant tokens in a sequence. This is a popular interpretation visualized in various Transformer architectures and settings ranging from encoder-decoder language translation (Bahdanau et al., 2014) to ViT image segmentation (Dosovitskiy et al., 2020; Caron et al., 2021). Mechanically, the softmax over query-key dot products exponentiates relative similarities between a query and each key, quantified via low-entropy or "spiky" attention weight distributions (Fig. 2).

Linear attention maps work by replacing the softmax with the normalized dot product of alternate feature maps (Eq. 2). With existing feature maps, we find the resulting attention weights can result in much higher entropy or more uniform distributions. This is true even for methods designed to approximate the softmax under mean-squared error bounds (Choromanski et al., 2020) (Performer, Fig. 2) or imposed locality (Qin et al., 2022b) (cosFormer, Fig. 2). This uniformity in attention weights reduces the modeling capacity of linear attentions leading to worse performance quality.

**Monotonicity over query-key dot products.** This property requires that the attention maps are monotonic over query-key dot products: when the dot product increases (decreases), the attention weight increases (decreases). In Fig. 3, we observe that while softmax attention exhibits this monotonicty (first subplot), the existing linear attentions do not. We believe this can cause training issues after swapping attentions due to conflicting gradients between attentions and original model parameters. In Fig. 3, trying to upweight attentions by increasing product similarity can actually result in

|  | BERT-FT | 1 + ELU | ReLU | Performer | cosFormer | exp(t = 1) | exp(t = 2) |
|---|---|---|---|---|---|---|---|
| Matthew's correlation | **58.8** | 28.1 | 39.5 | 24.7 | 39.9 | 45.9 | 50.0 |

Table 1: Finetuned-conversion performance of BERT finetuned on CoLA (BERT-FT), using prior linear attentions. With poor monotonicity (Fig. 3), prior methods fail to recover performance.

*decreased* attention weights. Later in Sec 3.2, we find this corresponds to failing to recover original performance when converting finetuned Transformers.

### 3.2 EXPLAINING THE LINEAR ATTENTION PERFORMANCE GAP

We validate the two properties introduced above by showing that (1) lacking spikiness corresponds to significantly worse performance when training from scratch, and (2) lacking spikiness and monotonicity corresponds to failing to recover performance when converting finetuned models.

**Training from scratch**. We compare various Transformers' abilities to solve Associative Recall (AR) (Ba et al., 2016), a next-token prediction task previously studied as a proxy for language modeling capability (Olsson et al., 2022). AR tests how well a model can recall specific content in an input sequence, structured as a list of key-value pairs which ends in a key (Table 12).

As a control for evaluating our hypothesis, we also consider a simple feature map designed to induce "spikiness" but not monotonicity: $\phi_t(x) = \exp(x \cdot t)$, which applies a temperature-$t$ scaled exponential element-wise.

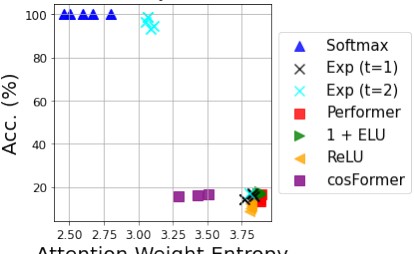

In Fig. 4, we observe a strong correspondence between low-entropy attention weights and AR accuracy. While softmax attention solves the AR task perfectly, prior linear attentions struggle to achieve even 20% accuracy, at the same time obtaining much larger attention weight entropies. As further support to our hypothesis, we see that while the exponential map $\phi_1$ fails AR and produces similarly high entropy attention weights, increasing spikiness with $t = 2$ actually solves the task.

Figure 4: Associative recall performance strongly corresponds to lower attention entropy; present in softmax attention but not prior linear variants.

**Finetuned-conversion**. We next compare how various linear attentions perform at recovering original softmax attention performance for finetuned-conversion. We adopt the procedure in Kasai et al. (2021), which takes a Transformer already finetuned on a specific task, swaps the attention layers with a linear attention variant, and further finetunes the entire model on the same task.

For this setting, we evaluate with a BERT-base-uncased model (Devlin et al., 2018) finetuned on the Corpus of Linguistic Acceptability (CoLA) task (Warstadt et al., 2019), where the goal is to classify whether a sentence is grammatically correct. We compare the performance of the original (softmax attention) BERT model[1] with the linear attention converted models. In Table 1, we find that just as no linear attention smoothly captures monotonicity over the trained model's query-key dot products, no linear attentions fully recover the original finetuned BERT's Matthew's correlation of 58.8. This includes the spiky $\phi_2$ feature map which was sufficient in the training-from-scratch regime.

## 4 HEDGEHOG: EXPRESSIVE LINEAR ATTENTION VIA SOFTMAX MIMICRY

We present Hedgehog, a simple, efficient, and expressive feature map trained to mimic softmax attention. Hedgehog is predicated by (1) there existing linear attention approximations to the softmax that recover the spiky and monotonic properties of standard attention in practice, and (2) that we can efficiently compute similar approximations efficiently.

In Sec. 4.1, we motivate Hedgehog and show that (1) is possible by revisiting low-degree Taylor polynomials. We find that for linear attention, the Taylor exponential works as a surprisingly simple feature map, recovering spikiness and monotonicity while matching standard Transformer performance. Unfortunately, we also find it introduces its own issues, where the feature map results in

---

[1]https://huggingface.co/JeremiahZ/bert-base-uncased-cola

large query and key dimensions and becomes inefficient to compute. In Sec. 4.2, to thus overcome these challenges, we propose and describe Hedgehog, a *trainable* linear attention trained to mimic softmax attention. In Sec. 5.1, we show how this enables similar spiky and monotonic properties to the softmax and Taylor exponential attentions, while retaining past linear attentions' efficiency.

## 4.1 SIMPLE POLYNOMIAL APPROXIMATIONS TO SOFTMAX ATTENTION

From our findings in Sec. 3, we seek an efficient linear alternative to the softmax which retains its spiky and monotonic properties. We first consider a simple potential approach: approximating the exponential in softmax by a low-degree Taylor polynomial (Keles et al., 2023; Banerjee et al., 2020).

While in general, a high-quality approximation to the softmax should retain its spiky, monotonic, and performant properties, we ground our investigation with two potential caveats for the Taylor polynomial. First, recall that feature maps for $p$-degree polynomial approximations can be computed in $\mathcal{O}(nd^p)$ time and space for every query and key vector. Thus, while this is indeed subquadratic in sequence length, the question remains whether we can set $p$ low enough to make the computation feasible while approximating $\exp$ reasonably. Second, as a general property of polynomials, the Taylor approximation only tracks its original function with low error in bounded regimes.

**Setup**. To test the Taylor approximation, we use the second-degree $\exp$ approximation, and evaluate on the prior train-from-scratch and finetuned-conversion settings (Sec. 3.2). We implement the feature map as $\exp(\boldsymbol{q}^\top \boldsymbol{k}) \approx \phi_{\text{taylor}}(\boldsymbol{q})^\top \phi_{\text{taylor}}(\boldsymbol{k})$, where $\phi_{\text{taylor}}(\boldsymbol{x})$ projects a $d$-dimensional query or key to $\mathcal{O}(d^2)$-dimensional features $\phi_{\text{taylor}}(\boldsymbol{x}) = \left[ 1, x_1, \ldots, x_d, \right] \cup \left[ x_i \cdot x_j \mid i, j \in [d] \right]$. **Positive results.** We find that the 2nd-degree Taylor approximation retains both the spikiness and monotonic properties (Fig. 5), and this corresponds to (near)-matching softmax attention performance (Table 2). We also note that here, the BERT query-key dot products are bounded in regimes where the second-order Taylor series $\exp$ approximation maintains monotonicity (Fig. 5). This suggests we can enable expressive linear attentions for training from scratch and finetuned-conversion.

**Caveats**. Unfortunately, the 2nd-degree Taylor approximation is not efficient. Even with $p = 2$, the feature map dimension is now $d' = 1 + d + d^2$, resulting in $\mathcal{O}(nd^3)$ attention complexity. As summarized in Table 2, this introduces an efficiency-effectiveness trade-off among functional attention approximations. Thus, the question remains whether we can recover the expressivity and modeling quality of softmax while achieving similar $\mathcal{O}(nd^2)$ scaling of past linear attentions.

## 4.2 LEARNABLE LINEAR ATTENTIONS FOR MIMICKING SOFTMAX

Our key insight is that rather than rely on fixed functional form that captures our spiky and monotonic properties, we can learn linear attention feature maps that do so. For each attention block, we propose feature maps as trainable single-layer MLPs, which is similar to prior work (Kasai et al., 2021) and acts similarly to an adapter (Houlsby et al., 2019) inserted after the query and key projections in Transformer attention layers (Fig. 1). However, unlike prior work, we explicitly train these feature maps such that the attention layers mimic the properties of softmax attention. We describe these two core components below, and validate these design choices in Sec. 5.1.

**Spiky MLP feature map.** Recall the kernel based linear attention paradigm from Sec. 2, where a feature map $\phi : \mathbb{R}^d \mapsto \mathbb{R}^{d'}$ is applied to both queries and keys to compute causal self-attention outputs using equation 2. However, unlike prior work that sticks to a pre-specified function as a feature map, we make the feature map a trainable MLP. In particular, for the single-head attention

| Method | Complexity | Spiky? | Mono-tonic? | Train-from-scratch (acc) | BERT-FT (MC) |
|---|---|---|---|---|---|
| Softmax | $\mathcal{O}(n^2 d)$ | ✓ | ✓ | 100.0 | 58.8 |
| 1 + ELU | $\mathcal{O}(nd'^2)$ | ✗ | ✗ | 17.0 | 28.1 |
| Performer | $\mathcal{O}(nd'^2)$ | ✗ | ✗ | 17.0 | 24.7 |
| CosFormer | $\mathcal{O}(nd^2)$ | ✗ | ✗ | 17.0 | 39.9 |
| Taylor Exp | $\mathcal{O}(nd^3)$ | ✓ | ✓ | **100.0** | **58.4** |

Table 2: Summary of feature maps compared to softmax, exhibiting an efficiency vs. expressivity tradeoff.

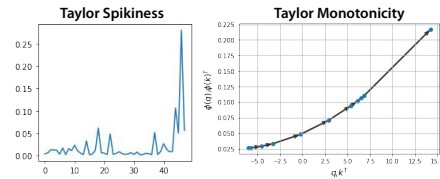

Figure 5: Taylor approximation recovers spikiness and monotonicity

setting, we compute $\phi_{\text{mlp}}(\boldsymbol{q}_i)^\top \phi_{\text{mlp}}(\boldsymbol{k}_j)$ with a simple one-layer MLP as $\phi_{\text{mlp}}(\boldsymbol{x}) = \Phi(\boldsymbol{W}^\top \boldsymbol{x} + \boldsymbol{b})$ where the matrix $\boldsymbol{W} \in \mathbb{R}^{d \times d'}$ and the bias $\boldsymbol{b} \in \mathbb{R}^{d'}$ are learned, and $\Phi$ is an activation function. To induce spikiness, we set $\Phi$ as the element-wise exponential function studied in Sec. 3.2, resulting in

$$\phi_{\text{mlp}}(\boldsymbol{x}) = \left[ \exp(\boldsymbol{w}_1^\top \boldsymbol{x} + \boldsymbol{b}), \ldots, \exp(\boldsymbol{w}_d^\top \boldsymbol{x} + \boldsymbol{b}) \right] \tag{3}$$

**Attention weight distillation loss.** To learn a softmax approximation, we train $\phi_{\text{mlp}}$ to minimize the cross-entropy loss between the computed linear attention weights and those that would have been computed via softmax attention. For query $\boldsymbol{q}_i$ and keys $\{\boldsymbol{k}_j\}_1^n$, we compute the sample losses as

$$\mathcal{L}_i = -\sum_{j=1}^i \frac{\exp(\boldsymbol{q}_i^\top \boldsymbol{k}_j)}{\sum_{m=1}^i \exp(\boldsymbol{q}_i^\top \boldsymbol{k}_m)} \log \frac{\phi_{\text{mlp}}(\boldsymbol{q}_i)^\top \phi_{\text{mlp}}(\boldsymbol{k}_j)}{\sum_{m=1}^i \phi_{\text{mlp}}(\boldsymbol{q}_i)^\top \phi_{\text{mlp}}(\boldsymbol{k}_j)} \tag{4}$$

For training Hedgehog attentions in multi-layer and multi-head attention Transformers, we apply a separate MLP to each head and each layer, and use the same $\phi_{\text{mlp}}$ for the queries and keys. We include further implementation details and pseudocode in Appendix A.

## 5 EXPERIMENTS

In experiments, we evaluate whether Hedgehog recovers softmax attention expressivity while retaining linear attention efficiency (Sec. 5.1), and how this improves modeling quality in training-from-scratch (Sec. 5.2), finetuned-conversion (Sec. 5.3), and pretrained-conversion regimes (Sec. 5.4).

### 5.1 BENCHMARKING HEDGEHOG FOR EXPRESSIVITY AND EFFICIENCY

Before evaluating Hedgehog on downstream tasks, we aim to validate Hedgehog's design choices for efficiency and expressivity. We address: (1) Do Hedgehog's spiky feature map and distillation loss recover the spiky and monotonic properties of softmax attention on the prior associative recall and BERT CoLA tasks? (2) Does Hedgehog achieve improved efficiency over softmax attention? (3) For conversion, do the learned attention weights actually match those of "ground-truth" softmax attention? Once learned, does this transfer to longer contexts and different tasks?

**Recovering softmax spiky and monotonic properties**. We test Hedgehog in the same train-from-scratch associative recall (AR) and finetuned-conversion of BERT on CoLA settings in Sec. 3.2. For training-from-scratch on AR, we do not use the distillation loss, and train the model end-to-end with next-token-prediction after inserting the learnable MLPs. In Table. 3, we find that Hedgehog achieves both favorable complexity and modeling for train-from-scratch and finetuned-conversion. This corresponds respectively with the spiky (Fig. 2) and monotonic (Fig. 3) properties noted prior.

**Recovering linear attention efficiency.** We next find Hedgehog's $\mathcal{O}(nd^2)$ scaling in compute and memory can lead to real-world efficiency gains. We benchmark inference in wall-clock time and memory usage for one attention layer with 12 heads and head dimension = 64 on sequences up to $n = 32K$ tokens long (Fig. 6). Hedgehog achieves near 6x faster inference and similar memory to FlashAttention (Dao et al., 2022) (linear in memory but quadratic in time). Meanwhile, the Taylor approximation, while $\mathcal{O}(n)$, gets significantly larger memory and slower speed due to the extra $d$.

**Recovering softmax attention weights**. We next study the *combination* of Hedgehog's feature map and distillation loss for matching softmax attention weights. Beyond recovering the spiky

| Method | Complexity | AR | BERT-FT |
|--------|-----------|-----|---------|
| Softmax | $\mathcal{O}(n^2d)$ | 100.0 | 58.8 |
| Taylor Exp | $\mathcal{O}(nd^3)$ | 100.0 | 58.4 |
| Hedgehog | $\mathcal{O}(nd^2)$ | **100.0** | **59.2** |

Table 3: Hedgehog matches performance on associative recall (AR) and BERT-finetuned conversion (BERT-FT) with prior best approaches, while achieving better time and space complexity.

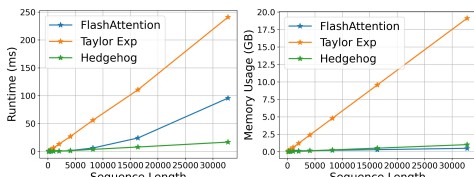

Figure 6: Hedgehog linear scaling in wall-clock time (left) and memory (right). Unlike the Taylor approx., Hedgehog inference gets real-world gains over FlashAttention.

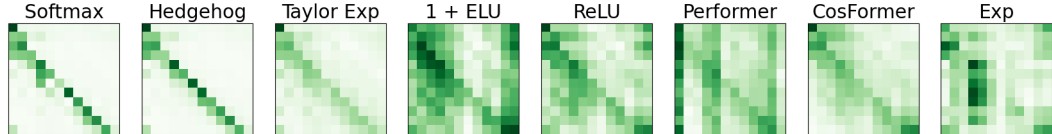

Figure 7: Compared to prior linear attentions, trained Hedgehog layers (**2nd left**) produce attention weights closely tracking softmax (**left**), with greater fidelity with both components (vs. Fig. 8).

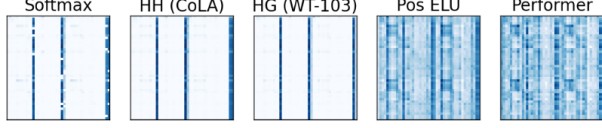

| Dataset | HH (CoLA) | HH (WT-103) | T2R-HH (CoLA) | HH (No Train) | 1 + ELU | Performer | CosFormer |
|---|---|---|---|---|---|---|---|
| CoLA | **0.172** | 0.352 | 0.191 | 0.694 | 1.223 | 1.293 | 1.196 |
| MRPC | 0.663 | **0.485** | 1.128 | 1.250 | 2.165 | 2.234 | 1.982 |
| MNLI | **0.345** | 0.382 | 0.613 | 0.851 | 1.51 | 1.581 | 1.338 |
| QNLI | 0.671 | **0.444** | 1.126 | 1.139 | 1.968 | 2.069 | 1.817 |

Figure 8: Hedgehog ablated attention weights.

Table 4: We find Hedgehog feature maps trained via distillation on CoLA or WikiText-103 generalize to new GLUE data, better matching softmax than prior linear attentions or ablations (reporting KL div.).

| Seq. Len | 256 | 1024 | 2048 | 4096 |
|---|---|---|---|---|
| CoLA KL | 0.182 | 0.187 | 0.190 | 0.181 |

Table 5: Hedgehog attention maintains fidelity with softmax attention over context lengths for BERT-FT on CoLA.

Figure 9: Hedgehog trained on CoLA and WT-103 recover softmax attentions on MRPC data.

and monotonic properties, learning to exactly match the weights can be particularly effective for converting or "distilling" pretrained quadratic Transformers into linear variants. For evaluation, we visualize the attention weights for different linear attentions in our BERT-FT CoLA setting (Fig. 7). We find Hedgehog recovers linear attention weights that match softmax's with much higher fidelity.

To further understand the contribution of Hedgehog's (1) spiky MLP and (2) distillation loss in Sec. 4.2, we visualize ablated attention weights by (1) using the distillation loss with the ReLU feature map used in Transformer-to-RNN (T2R-HH) (Kasai et al. (2021)), and (2) using untrained MLPs, replacing the trainable weights with an identity function (HH No Train). We find that distillation training is necessary to recover attention weights, and that the spiky MLP is also helpful for matching attentions (later supported by improved Transformer conversion in Sec. 5.3).

**Generalization to new data and longer contexts.** Finally, we investigate the generality of learned Hedgehog feature maps. We show Hedgehog attentions learned over specific data and context lengths can still better match softmax attention weights for new data and sequence lengths than prior linear attentions. We distill attentions for BERT models using CoLA or WikiText-103 (WT-103) samples, and report attention weights compared to softmax attention on three other GLUE tasks: qualitatively (Fig. 9) and quantitatively via KL divergence w.r.t. the "ground-truth" softmax weights (Table 4). We include additional visualizations and comparisons Appendix 5.

In Table 9, we further show that Hedgehog attention matching remains consistent across longer contexts. Post-distillation on CoLA samples, we concatenate CoLA samples into sequences 256 to 4096 tokens long (up to 8x the default 512 context length). We then compute attention weights using softmax and learned Hedgehog feature maps, and find that their KL divergence remains consistent.

## 5.2 LEARNING SEQUENCE MODELING FROM SCRATCH

We evaluate Hedgehog Transformers trained from scratch on the popular LRA sequence classification and WikiText-103 language modeling benchmarks. For training from scratch, we initialize MLPs as identity matrices for Hedgehog feature maps, and train the entire models end-to-end with the task-specific loss. We find Hedgehog achieves best average accuracy for both tasks among linear attentions (Table 6, 7). For LRA, while non-Transformer models are now state-of-the-art (Gu et al., 2021), our work focuses on approximating attention, so we compare with competitive subquadratic Transformers. We adopt the same hyperparameter settings as the official benchmark (Tay et al.,

| Model | ListOps | Text | Retrieval | Image | Pathfinder | Average |
|---|---|---|---|---|---|---|
| Transformer | 36.37 | 64.27 | 57.46 | 42.44 | 71.40 | 54.39 |
| Local Att | 15.82 | 52.98 | 53.39 | 41.46 | 66.63 | 46.06 |
| Linear Trans. | 16.13 | **65.90** | 53.09 | 42.34 | 75.30 | 50.55 |
| Reformer | 37.27 | 56.10 | 53.40 | 38.07 | 68.50 | 50.67 |
| Sparse Trans. | 17.07 | 63.58 | 59.59 | 44.24 | 71.71 | 51.24 |
| Sinkhorn Trans. | 33.67 | 61.20 | 53.83 | 41.23 | 67.45 | 51.29 |
| Linformer | 35.70 | 53.94 | 52.27 | 38.56 | 76.34 | 51.36 |
| Performer | 18.01 | 65.40 | 53.82 | 42.77 | **77.05** | 51.41 |
| Synthesizer | 36.99 | 61.68 | 54.67 | 41.61 | 69.45 | 52.88 |
| Longformer | 35.63 | 62.85 | 56.89 | 42.22 | 69.71 | 53.46 |
| BigBird | 36.05 | 64.02 | 59.29 | 40.83 | 74.87 | 55.01 |
| Nystrmformer[†] | 37.15 | 65.52 | 79.56 | 41.58 | 70.94 | 58.95 |
| cosFormer[†] | 37.90 | 63.41 | 61.36 | **43.17** | 70.33 | 55.23 |
| Skyformer[†] | **39.25** | 64.70 | 82.06 | 40.77 | 70.73 | 59.50 |
| Hedgehog | 37.15 | 64.60 | **82.24** | 40.15 | 74.16 | **59.66** |

Table 6: Training-from-scratch on LRA. Hedgehog achieves best avg. acc. (%) across most competitive Transformers (full results in Table 13, trends hold). [†] indicates results reported from original works. All others reported from the official LRA benchmark (Tay et al., 2021). **Best**, 2nd-best.

| Model | Transformer | Performer | Reformer | AFT | (1 + ELU) | Hedgehog |
|---|---|---|---|---|---|---|
| Perplexity | **18.6** | 26.8 | 25.6 | 28.2 | 25.6 | 20.8 |

Table 7: Training-from-scratch on WikiText-103. Among 125M decoder-only models, Hedgehog significantly closes the gap between standard Transformers and prior linear attention maps by 68.6%.

2021). On WikiText-103, we adopt the setting in Fu et al. (2023), evaluating a 125M decoder-only Transformer on perplexity over 1024 tokens. Hedgehog significantly closes the gap by up to 6 PPL.

## 5.3 FINETUNED CONVERSION OF QUADRATIC TO LINEAR TRANSFORMERS

For the finetuned Transformer conversion regime, we evaluate performance recovery for BERT-base models finetuned on GLUE, and ViT-B/16 models trained on ImageNet-1K. For both settings, we first swap attentions and train via our distillation loss (Sec. 4.2). We then finetune the converted BERT models on their original tasks as in Transformer-to-RNN (T2R) (Kasai et al., 2021).

For BERT, we compare Hedgehog to T2R in Table 8, and find that in contrast, Hedgehog conversion recovers near-100% of the original softmax attention performance. To further test Hedgehog's feature map and attention distillation, we also compare against an ablation that trains the T2R feature map with our distillation loss (T2R-HH). We find that training to mimic softmax attentions boosts performance of T2R, suggesting that attention weight distillation may be a general step to improving linear attention feature maps. However, Hedgehog's exponential still leads to superior performance. We find similar results for ViT-B/16, suggesting Hedgehog can also apply to other modalities.

## 5.4 PRETRAINED CONVERSION FOR SUBQUADRATIC TASK TRANSFER

We finally evaluate Hedgehog for converting pretrained Transformers into linear Transformers. We consider two settings: (1) To benchmark Hedgehog and the pretrained-conversion regime for subquadratic sequence modeling, we use the same WT-103 evaluation in Sec. 5.2 for converting 125M-parameter GPT-2. (2) As an early application for Hedgehog on larger models, we convert Llama-2 7B (Touvron et al., 2023) before finetuning with low-rank adapters (LoRA) (Hu et al., 2021) on SAMSum summarization (Gliwa et al., 2019). We include further training details in Appendix. B.5.

To most directly measure pretrained-conversion quality, for both settings we compare against T2R. For GPT-2, we find Hedgehog both outperforms T2R, and further outperforms modern subquadratic sequence models such as H3 (Fu et al., 2023) and Hyena (Poli et al., 2023) (Table 10). Although not directly comparable due to pretraining, we also compare with zero-shot and finetuned GPT-2 for reference. While Hedgehog is 1 PPL off the fully quadratic finetuned GPT-2, it significantly

improves over zero-shot while being linear to train. We finally apply Hedgehog for Llama-2 conversion, where Hedgehog enables linear attention Llamas that train via LoRA (see Appendix C.3 for sample generations).

| Method | CoLA | SST2 | MRPC | STS-B | QQP | MNLI | QNLI | RTE | (%) Recover |
|---|---|---|---|---|---|---|---|---|---|
| BERT-FT | 58.8 | 93.2 | 90.2 | 88.8 | 91.0 | 84.7 | 91.3 | 68.2 | 100.0 |
| T2R | 43.6 | 87.7 | 83.0 | 78.6 | 86.7 | 78.9 | 84.6 | 54.1 | 88.9 |
| T2R-HH | 56.9 | 90.9 | 89.1 | 77.7 | 90.0 | 77.4 | 84.5 | 56.3 | 93.5 |
| Hedgehog | **59.2** | **92.6** | **90.1** | **87.4** | **91.0** | 82.6 | **89.6** | **69.3** | **99.3** |

Table 8: Finetuned-conversion evaluation. Hedgehog recovers 99.3% of original finetuned BERT (BERT-FT) GLUE performance.

| Top-1 | Acc. % |
|---|---|
| ViT-B/16 | 80.3 |
| T2R-HH | 77.0 |
| Hedgehog | **79.5** |

Table 9: Hedgehog achieve 99% ViT acc.

| Method | GPT-2 | GPT-2 FT | Hybrid H3 | Hyena | T2R-GPT-2 | HH-GPT-2 |
|---|---|---|---|---|---|---|
| PPL | 28.0 | 15.8 | 18.5 | 18.5 | 19.4 | **16.7** |

Table 10: Pretrained-conversion for 125M GPT-2 on WT-103 lang. modeling. While finetuned GPT-2 gets lowest PPL, among *subquadratic* models Hedgehog significantly outperforms by 1.8 PPL.

| Llama-2 | R1 / R2 / RL |
|---|---|
| Softmax (Zero-shot) | 19.3 / 6.8 / 14.9 |
| Softmax (LoRA) | 51.1 / 27.6 / 43.5 |
| T2R (LoRA) | 2.8 / 0.0 / 2.6 |
| Hedgehog (LoRA) | **47.4 / 23.4 / 39.1** |

Table 11: Hedgehog Llama-2 conversion (ROUGE).

## 6 CONCLUSION

We present Hedgehog, a learnable linear attention to mimic softmax attention. This enables training linear attention models from scratch and *converting* existing Transformers into linear attention variants. To motivate Hedgehog we study why prior linear attentions underperform softmax attention, and identify two missing properties: (1) the ability to capture low entropy or spiky attention maps and (2) to be monotonic with respect to the underlying query-key dot products. We find training to match softmax attentions results in recovering many of its expressive properties, and that Hedgehog leads to competitive performance with softmax-based attention in training from scratch, finetuned-conversion, and pretrained conversion regimes.

## ACKNOWLEDGEMENTS

We thank Armin Thomas, Gordon Downs, Krista Opsahl-Ong, Pun Waiwitlikhit, Schwinn Saereesitthipitak, Dan Fu, Simran Arora, Sabri Eyuboglu, Tri Dao, and anonymous reviewers for helpful discussions on linear attention and paper feedback. We also thank Dan Fu for prior versions of the pseudocode formatting in the appendix.

We gratefully acknowledge the support of NIH under No. U54EB020405 (Mobilize), NSF under Nos. CCF1763315 (Beyond Sparsity), CCF1563078 (Volume to Velocity), and 1937301 (RTML); US DEVCOM ARL under No. W911NF-21-2-0251 (Interactive Human-AI Teaming); ONR under No. N000141712266 (Unifying Weak Supervision); ONR N00014-20-1-2480: Understanding and Applying Non-Euclidean Geometry in Machine Learning; N000142012275 (NEPTUNE); NXP, Xilinx, LETI-CEA, Intel, IBM, Microsoft, NEC, Toshiba, TSMC, ARM, Hitachi, BASF, Accenture, Ericsson, Qualcomm, Analog Devices, Google Cloud, Salesforce, Total, the HAI-GCP Cloud Credits for Research program, the Stanford Data Science Initiative (SDSI), and members of the Stanford DAWN project: Meta, Google, and VMWare. The U.S. Government is authorized to reproduce and distribute reprints for Governmental purposes notwithstanding any copyright notation thereon. Any opinions, findings, and conclusions or recommendations expressed in this material are those of the authors and do not necessarily reflect the views, policies, or endorsements, either expressed or implied, of NIH, ONR, or the U.S. Government.

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

## A  HEDGEHOG IMPLEMENTATION DETAILS

We provide further details on the Hedgehog feature map and attention weight distillation training.

### A.1  MECHANICS FOR HEDGEHOG FEATURE MAP

To improve Hedgehog performance in practice, we explored variations along two additional criteria for numerical stability and improved expressivity.

**Numerical stability**  In practice, we find that computing $\Phi$ as the softmax applied over the *MLP output dimension* also seems to work but with better stability. In this case, we expand Eq. 3 as

$$\phi_{\mathrm{mlp}}(\boldsymbol{x}) = \Big[ \frac{\exp(\boldsymbol{w}_1^\top \boldsymbol{x})}{\sum_{i=1}^d \exp(\boldsymbol{w}_i^\top \boldsymbol{x})}, \cdots, \frac{\exp(\boldsymbol{w}_d^\top \boldsymbol{x})}{\sum_{i=1}^d \exp(\boldsymbol{w}_i^\top \boldsymbol{x})} \Big] \tag{5}$$

(also performing better than dividing each element by the max over $\{\exp(\boldsymbol{w}_i^\top \boldsymbol{x} + \boldsymbol{b})\}_{i=1}^d$)

**Negation mapping**.  To better compute dot products as a similarity measure between queries and keys, in practice we also set $\Phi$ as a mapping from $\mathbb{R}^d \mapsto \mathbb{R}^{2d}$, *e.g.,* via

$$\phi_{\mathrm{mlp}}(\boldsymbol{x}) = \Big[ \exp(\boldsymbol{w}_1^\top \boldsymbol{x} + \boldsymbol{b}), \ldots, \exp(\boldsymbol{w}_d^\top \boldsymbol{x} + \boldsymbol{b}), \exp(-\boldsymbol{w}_1^\top \boldsymbol{x} - \boldsymbol{b}), \ldots, \exp(-\boldsymbol{w}_d^\top \boldsymbol{x} - \boldsymbol{b}) \Big] \tag{6}$$

where the additional negation mapping in $\mathbb{R}^{2d}$ intuitively lets us better factor in negative dimensionalities, which prior linear attention feature maps like ReLU ignore. While this results in a larger feature dimension, it only scales by a fixed constant 2 such that the overall time and space complexity for Hedgehog linear attention is still $\mathcal{O}(nd^2)$. We further find that in practice, this still accomplishes favorable scaling and much faster inference with smaller memory than the Taylor exponential discussed in Sec. 4.1 (see Fig. 6 for real-world wall-clock time and memory savings).

### A.2  HEDGEHOG FEATURE MAP AND MODEL ARCHITECTURE

We apply Hedgehog feature maps for each head and layer individually in a standard Transformer architecture, where the addition of head-specific MLPs is akin to inserting "adapters" (Houlsby et al., 2019) after every query and key projection. Each MLP is a single linear layer with input and output dimensions equal to the base Transformer's head dimension. Pytorch-like code is given below.

```python
import torch
import torch.nn as nn

class HedgehogFeatureMap(nn.Module):
    def __init__(self, head_dim: int, activation: str = 'exp'):
        super().__init__()
        # Trainable map
        self.layer = nn.Linear(head_dim, head_dim)
        self.init_weights_()

    def self.init_weights_(self):
        """Initialize trainable map as identity"""
        nn.init.eye_(self.layer.weight)
        nn.init.zeros_(self.layer.bias)

    def forward(self, x: torch.Tensor):
        x = self.layer(x)   # shape b, h, l, d
        return torch.cat([torch.exp(x), torch.exp(-x)], dim=-1)
```

### A.3  HEDGEHOG DISTILLATION AND FINETUNING IMPLEMENTATION DETAILS

We include additional details for training Hedgehog layers to obtain linear attention Transformers. These fall under two categories: (1) training-from-scratch, and (2) finetuned / pretrained conversion.

**1. Training-from-scratch**. When training Hedgehog Transformers from scratch, we insert a Hedgehog MLP for each query and key projection of the randomly initialized Transformer (*e.g.,* for each head of a multi-head attention layer, and for all such layers). We then train the Hedgehog MLPS jointly with all other model parameters end-to-end with a single objective function, *e.g.,* cross-entropy loss on next-token prediction when training models for language modeling.

**2. Finetuned / pretrained conversion**. For both these regimes, we carry out training as a two stage process. Like training-from-scratch, we initially insert Hedgehog MLPs for query and key projections. Following this, we proceed in two stages:

1. **Attention distillation.** We first freeze the Transformer's original weights and specifically train the Hedgehog MLPs, such that the resulting linear attention weights match those produced via softmax attention over the same query and key tensors. For each head, we conceptually follow Listing 1 below to compute a soft cross-entropy or KL-divergence between the "predicted" linear attention weights and "ground-truth" softmax attention weights. We compute these losses for each attention head and layer after one forward pass of the entire model, using data samples from the target task. We find it sufficient to use one optimizer for joint training over all Hedgehog layers in parallel, using the sum of each individual attention head distillation loss as the final criterion. This makes training simple and comparable to a standard training loop; we further provide code[†] in Listing 2 to do so with popular APIs such as HuggingFace Transformers[2].

2. **Original parameter finetuning.** Following attention distillation, we simply unfreeze all model weights and train with a standard task-specific loss function. We find we can also keep certain layers frozen or train with parameter-efficient finetuning such as low-rank adaptation (Hu et al., 2021); we explore this in Sec. 5.4 with Llama-2 models.

```python
# Hedgehog distillation loss for one attention head

def softmax_attn(q: torch.Tensor, k: torch.Tensor):
    """Get softmax attention weights -> Assume q, k are both shape (b, h,
    l, d)"""
    scale = q.shape[-1] ** 0.5
    qk = torch.einsum('bhmd,bhnd->bhmn', q, k) / scale
    return torch.softmax(qk, dim=-1)

def quadratic_linear_attn(q: torch.Tensor, k: torch.Tensor):
    """
    Get linear attention weights
    -> Assume q, k are both shape (b, h, l, d), and feature maps already
    applied
    """
    qk = torch.einsum('bhmd,bhnd->bhmn', q, k)
    return qk / qk.sum(dim=-1, keepdim=True)

def compute_hedgehog_loss(q: torch.Tensor,
                          k: torch.Tensor,
                          hh_mlp_q: HedgehogFeatureMap,
                          hh_mlp_k: HedgehogFeatureMap):
    """
    Compute the attention distillation loss
    -> Assume `soft_label_cross_entropy` is implemented
        (alternatively use KL divergence)
    -> Assume q and k are the queries and keys of a
        pretrained Transformer,
        e.g., via q = self.q_proj(hidden_states)
    """
    true_attn = softmax_attn(q, k)
    pred_attn = quadratic_linear_attn(hh_mlp_q(q), hh_mlp_k(k))
    return soft_label_cross_entropy(pred_attn, true_attn)
```

Listing 1: Hedgehog distillation loss for one attention head

[2]https://huggingface.co/docs/transformers/index

```python
# Hedgehog Attention class for easy attention distillation

class HedgehogAttention(nn.Module):
    """
    Sample code for HedgehogAttention, following HuggingFace API
    """
    def __init__(self, base_attn, training = True):
        self.base_attn = base_attn  # e.g., LlamaAttention

        # Trainable feature maps
        self.mlp_q = HedgehogFeatureMap(base_attn.head_dim)
        self.mlp_k = HedgehogFeatureMap(base_attn.head_dim)

        # Freeze original attention parameters
        for p in self.base_attn.parameters():
            p.requires_grad = False

        self.q_proj = self.base_attn.q_proj
        self.k_proj = self.base_attn.k_proj

        # Whether we train attentions or not
        self.training = training

    def forward(self,
                hidden_states: torch.Tensor,
                output_attentions: bool = True,
                **base_kwargs: any):

        if self.training:
            # Compute ground-truth attention weights
            outputs, true_attns = self.base_attn(
                hidden_states=hidden_states,
                output_attentions=True,
                **base_kwargs)

            # Compute Hedghog feature maps
            q = self.mlp_q(self.q_proj(hidden_states))
            k = self.mlp_k(self.k_proj(hidden_states))

            pred_attns = quadratic_linear_attn(q, k)

            if output_attentions:  # Hook for attentions
                return outputs, (pred_attns, true_attns)

        # ... End relevant
```

Listing 2: Hedgehog Attention class for easy attention distillation.

[†]In practice, to train all Hedgehog layers easily in a joint end-to-end fashion, we make use of popular pretrained Transformer APIs such as those in the HuggingFace transformers library. We implement a Hedgehog equivalent of the base Transformers' attention class, which (1) abstracts away the Transformer-specific attention computation and (2) lets us hook attention weights calculated at each layer to the model's final outputs, *e.g.,* via output_attentions = True keyword args. We can subsequently substitute each attention layer with the "HedgehogAttention" equivalent, and train via a simple loop over the data. We present Pytorch-like code in Listing 3.

```python
# End-to-end joint attention distillation

from transformers import AutoModel

# Load base model
base_model = AutoModel.from_pretrained(...)

# Freeze original parameters
for p in base_model: p.requires_grad = False

# Convert attentions for all layers
for layer in base_model:
    base_model.attn = HedgehogAttention(base_model.attn)

# Define single optimizer for training all feature maps
optim = optimizer(base_model.parameters())

# Train Hedgehog feature maps
for data in dataloader:

    # Compute outputs and hook to attentions
    outputs = base_model(**data, output_attentions=True)
    outputs = outputs.get('attentions')

    total_loss = 0
    for attns in enumerate(outputs):  # attentions for each layer
        pred_attn, true_attn = attns
        total_loss += soft_label_cross_etnropy(pred_attn, true_attn)

    loss.backward()  # Jointly optimize all feature maps
    optim.step()
```

Listing 3: End-to-end joint attention distillation.

## B DEFERRED EXPERIMENTAL DETAILS

### B.1 ASSOCIATIVE RECALL ANALYSIS (SECTION 3.2)

In Sec. 3.2, we compare various Transformers' abilities to solve Associative Recall (AR) (Ba et al., 2016), a next-token prediction task previously studied as a proxy for language modeling capability (Olsson et al., 2022). AR tests how well a model can recall specific content in an input sequence, structured as a list of key-value pairs which ends in a key Table 12.

| Input Sequence | Next Token | Vocab Size | Seq. Length |
|---|---|---|---|
| c 9 k 8 j 3 ... f 1 c | 9 | 40 | 128 |

Table 12: Associative recall task. Example from Ba et al. (2016).

**Dataset details.** To understand the effects of more uniform attention weightings, we evaluate with 40 possible tokens and 128 token-long-sequences, such that models must recall pairings that only occur three times on average in-context. We generate 10,000 training samples following the patterns described in Table 12, and evaluate on 2000 newly-generated test samples (again using the same associative recall structure, but with different token associations).

**Architecture details.** For all experiements, we use a four layer Transformer with four heads-per-layer, head dimension = 64, and rotary embeddings. This is similar to modern model families such

as Pythia (Biderman et al., 2023) and LLaMA / Llama-2 (Touvron et al., 2023). We keep all parts consistent except for the multi-head attention, comparing popular linear attentions (c.f., Fig. 2).

**Training details.** For fair comparison to evaluate just the feature map / modeling architecture, we train all models by sweeping learning rate $\in \{$1e-2, 1e-4$\}$, weight decay $\in \{$0, 5e-4$\}$, and batch size $\in \{$8, 32$\}$ with AdamW optimizer. We train up to 100 epochs with early stopping (explicitly stopping training if validation loss stops decreasing after 10 epochs).

### B.2 BERT-BASE FINETUNED ON COLA CONVERSION (SECTION 3.2)

**Training details.** For our finetuned-conversion analysis, we replace the attentions of a finetuned BERT-base-uncased model available on the HuggingFace model hub[3]. We train with batch size 8, learning rate 1e-5, zero weight decay, AdamW optimizer, and up to 10 epochs with early stopping.

### B.3 HEDGEHOG TRAINING FROM SCRATCH (SECTION 5.2)

**LRA training and model details.** On LRA, for fair comparison we implement Hedgehog in the existing PyTorch implementation provided by Xiong et al. (2021), deferring to the same model configurations and hyperparameters used in the original repository (Tay et al., 2021).

**WikiText-103 training and model details.** For WikiText-103, we train a 125M parameter GPT-2 style Transformer with learning rate 6e-4, weight decay 0.01, and AdamW optimizer. For close comparison, we follow the architectural details of GPT-2 125M, and use a 12 layer decoder-only network with 12 heads, head dimension = 64, hidden dimension 768, and MLP dimension 3072.

### B.4 HEDGEHOG FINETUNED CONVERSION (SECTION 5.3)

**Recovering finetuned BERT performance on GLUE tasks.** For finetuned conversion, we first conduct Hedgehog attention distillation by training attention layers up to five epochs with early stopping based on validation loss. We train with learning rate 1e-2, weight decay 0, AdamW optimizer. We follow the same procedure for the Transformer-to-RNN (T2R) (Kasai et al., 2021) ablation. For regulard T2R and subsequently post attention distillation, we train each BERT model with batch size 8, learning rate 1e-5, weight decay 0, AdamW optimizer, and cosine scheduler for up to five epochs on the individual classification (all except STS-B) or regression tasks (STS-B) on the GLUE benchmark. For all tasks, we use the corresponding available finetuned BERT-base-uncased checkpoints hosted at the HuggingFace model hub[4], and thank the original uploader for their contributions.

**Recovering finetuned Vision Transformer performance on ImageNet-1K.** To demonstrate finetuned-conversion for the image domain, we use the `vit-base-patch16-224` checkpoint provided by Google on HuggingFace[5], which is trained on ImageNet-21k before being finetuned on ImageNet-1K at resolution of 224 x 224 pixels (Dosovitskiy et al., 2020). For distillation, we freeze the original ViT weights, and train linear attention MLPS with batch size 32, learning rate 0.01, zero weight decay, and AdamW optimizer, and train for two epochs. We then train all parameters with learning rate 1e-3, zero weight decay and AdamW optimizer up to 10 epochs with early stopping.

### B.5 HEDGEHOG PRETRAINED CONVERSION (SECTION 5.4)

**Linear GPT-2 125M conversion for WikiText-103 language modeling.** We use the available GPT-2 125M pretrained checkpoint available on HuggingFace[6] from Radford et al. (2019). For Hedgehog, we first do attention distillation and train Hedgehog MLPs for two epochs over the WikiText-103 data, using batch size 8, learning rate 0.01, zero weight decay, AdamW optimizer, and 1024-tokens per input. For T2R-GPT-2 and the subsequent Hedgehog-GPT-2 model, we finetune all model parameters with learning rate 6e-4, weight decay 0.01, and AdamW optimizer and 1024 tokens-per-input.

---

[3]https://huggingface.co/JeremiahZ/bert-base-uncased-cola

[4]https://huggingface.co/JeremiahZ/

[5]https://huggingface.co/google/vit-base-patch16-224

[6]https://huggingface.co/gpt2

**Linear Llama-2 7B conversion for SAMSum corpus summarization.** We use the base Llama-2 7B model available via Meta and HuggingFace (`llama-2-7b-hf`) from Touvron et al. (2023). For all experiments, we use non-quantized model weights in bfloat16, and conduct all training runs and evaluations on a single A6000 GPU.

For dataset preparation, we first convert individual document and summarization pairs into single next-token prediction samples, using the template in Listing 4. For both distillation and subsequent finetuning, we then chunk these samples into concatenated inputs 1024 tokens long.

For attention distillation, we freeze all original Llama-2 weights, and train Hedgehog MLPs for every head and layer (0.495% of the original model size). We then train for two epochs with learning rate 0.01, zero weight decay, AdamW optimizer, and batch size 8 with gradient accumulation.

For finetuning and comparison to T2R and standard attention, we apply LoRA to query, key, value, and output projections of each layer. We use alpha parameter 16 and rank 8. We train with learning rate 1e-4, zero weight decay, AdamW optimizer, and batch size 8 with gradient accumulation.

For generation, we compute ROUGE metrics (R1, R2, RL; for overlap of unigrams, bigrams, and longest common subsequence) over model outputs. We generate sequences up to 100 tokens long, and evaluate based on outputs up til the first `` Llama stop token.

```
1  # Llama-2 prompt template for SAMSum
2
3  Summarize this dialog:
4  {input}
5  ---
6  Summary:
7  {output}{eos_token}
```

Listing 4: Llama-2 prompt template for SAMSum corpus summarization

## C ADDITIONAL RESULTS

### C.1 EXTENDED COMPARISON TO ATTENTION MODELS ON LRA

In Table 13, we compare Hedgehog's performance on LRA against a fuller set of Transformer and subquadratic Transformer based alternatives sourced either from the official benchmark leaderboard (Tay et al., 2021) or recent subquadratic attention works (where we display the most competitive alternatives in Table 6). We find Hedgehog on average obtains best accuracy. Although recently non-Transformer models such as deep state-space models have shown impressive results outperforming Transformers on LRA (Gu et al., 2021), as our work focuses on how to improve and recover the expressivity of standard softmax Transformers, we focus the comparison against other attention-based methods. We defer to Gu et al. (2021) and related works for their LRA results.

### C.2 HEDGEHOG FEATURE MAP GENERALIZATION TO NEW DATA

We extend our analysis into how Hedgehog's feature maps learned with one dataset generalize to attentions computed on a new dataset (*c.f.* Table 4 and Fig. 9 in Sec. 5.1). As in the prior section, we find Hedgehog learned feature maps frequently generalize to new datasets. Despite training to match the softmax attentions on one model and dataset, we first find Hedgehog feature maps can produce attention weights that closely resemble softmax attention for the same model on another dataset (App. C.2.1). We next quantify this fidelity via KL divergence w.r.t. the softmax attentions (App. C.2.2). We find that Hedgehog learned feature maps almost always still generalize better than prior linear attention feature maps. We finally show that this attention matching generalization transfers to actual pretrained-conversion performance (App. C.2.3). We replace BERT-base softmax attentions with Hedgehog attentions trained on one task, and find finetuning with these converted models on *another* GLUE task still leads to improvements over prior linear attentions.

**Setup.** For all experiments, we begin by training Hedgehog attentions on "in-distribution" softmax attention data. We use the pretrained BERT-base-uncased model (Devlin et al., 2018) as the Trans-

| Model | ListOps | Text | Retrieval | Image | Pathfinder | Average |
|---|---|---|---|---|---|---|
| Transformer | 36.37 | 64.27 | 57.46 | 42.44 | 71.40 | 54.39 |
| Local Att | 15.82 | 52.98 | 53.39 | 41.46 | 66.63 | 46.06 |
| Linear Trans. | 16.13 | 65.90 | 53.09 | 42.34 | 75.30 | 50.55 |
| Reformer | 37.27 | 56.10 | 53.40 | 38.07 | 68.50 | 50.67 |
| Sparse Trans. | 17.07 | 63.58 | 59.59 | 44.24 | 71.71 | 51.24 |
| Sinkhorn Trans. | 33.67 | 61.20 | 53.83 | 41.23 | 67.45 | 51.29 |
| Linformer | 35.70 | 53.94 | 52.27 | 38.56 | 76.34 | 51.36 |
| Performer | 18.01 | 65.40 | 53.82 | **42.77** | **77.05** | 51.41 |
| Synthesizer | 36.99 | 61.68 | 54.67 | 41.61 | 69.45 | 52.88 |
| Longformer | 35.63 | 62.85 | 56.89 | 42.22 | 69.71 | 53.46 |
| BigBird | 36.05 | 64.02 | 59.29 | 40.83 | 74.87 | 55.01 |
| Nystrmformer[†] | 37.15 | 65.52 | 79.56 | 41.58 | 70.94 | 58.95 |
| cosFormer[†] | 37.90 | 63.41 | 61.36 | 43.17 | 70.33 | 55.23 |
| Skyformer[†] | **39.25** | 64.70 | 82.06 | 40.77 | 70.73 | 59.50 |
| Hedgehog | 37.15 | 64.60 | **82.24** | 40.15 | 74.16 | **59.66** |

Table 13: Training-from-scratch on LRA. Hedgehog achieves best average performance across Transformers and subquadratic variants. [†] indicates method results reported from original works. All other reported from the official LRA benchmark (Tay et al., 2021). **Best**, 2nd-best acc (%).

former we wish to convert, and distill two sets of Hedgehog attentions over (1) the GLUE CoLA task or (2) 512-token chunks of WikiText-103 corpus. Thus, queries and keys computed with the BERT-base-uncased model over CoLA validation samples are "in-distribution" for the first set, and we are interested in seeing how attention weight fidelity or downstream performance recovery are affected when subsequently finetuning on non-CoLA GLUE data. We compare with various prior ablations and alternative feature maps, such as the Transformer-to-RNN feature map (Kasai et al., 2021) after attention distillation, Hedgehog without attention distillation, and prior representative linear attentions such as Performer (Choromanski et al., 2020) and cosFormer (Qin et al., 2022b).

### C.2.1 QUALITATIVE EVIDENCE OF HEDGEHOG DATA GENERALIZATION

In Fig. 10 and Fig. 11, we visualize attention weights computed via various methods on heads in the 1st, 6th, and 12th layers of the BERT-base uncased model. We find Hedgehog can learn feature maps that lead to matching softmax attention weights, even when computed on new data samples. Interestingly, the Hedgehog feature maps result in significantly more similar attention weights versus alternative feature maps (quantified in the next section).

In addition, our comparisons to Hedgehog ablations suggest that the proposed Hedgehog feature map *and* distillation procedure are important for best generalization. Removing either the Hedgehog feature map form (via doing attention distillation using the prior Transformer-to-RNN feature map (T2R-HH) or not training feature maps (HH (No Train)) leads to lower fidelity, where attention distillation seems critical for retaining weights reasonably similar to softmax attention.

### C.2.2 QUANTITATIVE ANALYSIS OF HEDGEHOG DATA GENERALIZATION

To quantify the above observations, we compute the KL divergence between Hedgehog attention weights computed on various GLUE tasks and the "ground-truth" softmax attention weights, using the pretrained BERT-base-uncased model. We report the KL divergence in Table 14. Similar to the above visualizations, we find that Hedgehog feature maps do seem to produce better matching attention weights to softmax attention via significantly smaller KL divergences.

### C.2.3 HEDGEHOG DATA GENERALIZATION VIA GLUE TASK TRANSFER

We finally evaluate the Hedgehog attention generalization by finetuning the pretrained BERT models with trained Hedgehog on new GLUE tasks. We follow the same procedure described in Ap-

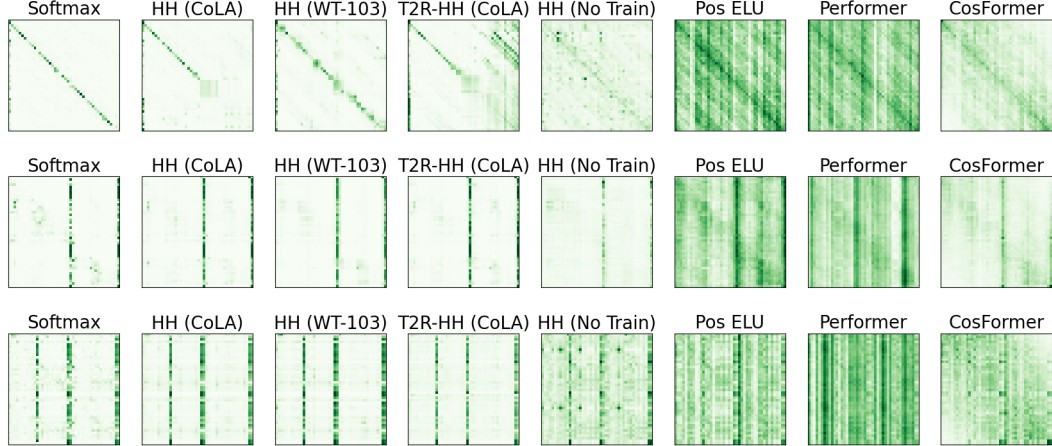

Figure 10: **Qualitative generalization to MRPC**. Attention weights for BERT-base-uncased queries and keys computed on MRPC samples. We compare attentions from the 3rd head in the 1st, 6th and 12th layers (top, middle, bottom). Hedgehog feature maps trained on CoLA or WikiText-103 often still produce attention weights similar to those of softmax attention on new data.

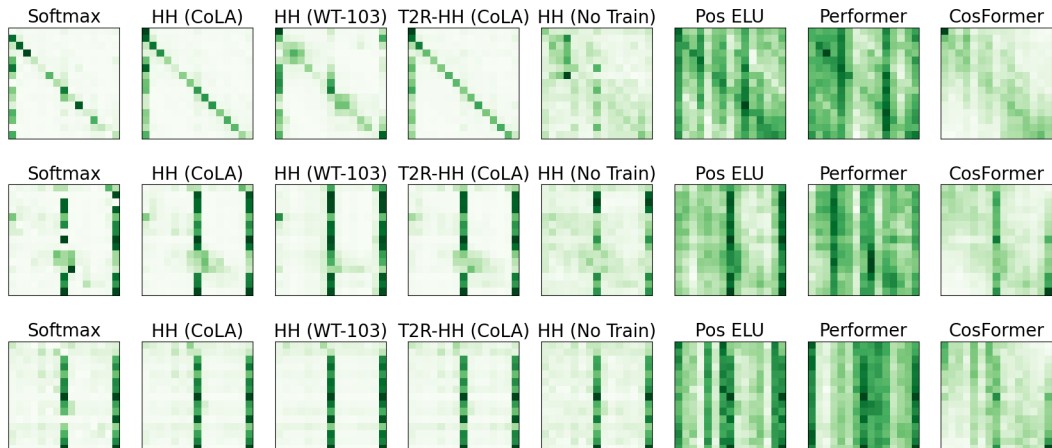

Figure 11: **Qualitative generalization to QNLI**. Attention weights for BERT-base-uncased queries and keys computed on QNLI samples. We compare attentions from the 3rd head in the 1st, 6th and 12th layers (top, middle, bottom). Hedgehog feature maps trained on CoLA or WikiText-103 often still produce attention weights similar to those of softmax attention on new data.

pendix B.4. In Table 15, we find that the above attention weight observations on Hedgehog generalization also correspond with downstream task performance. Hedgehog-BERT models achieve best or second-best performance, despite using attention feature maps trained on different data. We leave further generalization studies, such as how Hedgehog attentions trained on one model generalize to *an entirely different* model for future work.

| Method | CoLA | MNLI | MRPC | QNLI | QQP | RTE | SST-2 | STS-B |
|---|---|---|---|---|---|---|---|---|
| Hedgehog (CoLA) | **0.173** | **0.340** | 0.652 | 0.673 | **0.374** | 0.932 | **0.211** | **0.275** |
| Hedgehog (WT-103) | 0.357 | 0.381 | **0.484** | **0.446** | 0.432 | **0.428** | 0.347 | 0.360 |
| T2R-HH (CoLA) | 0.191 | 0.603 | 1.124 | 1.141 | 0.658 | 1.357 | 0.254 | 0.444 |
| Hedgehog Untrained | 0.687 | 0.845 | 1.264 | 1.146 | 0.890 | 1.493 | 0.859 | 0.743 |
| 1 + ELU | 1.231 | 1.500 | 2.150 | 1.947 | 1.509 | 2.491 | 1.505 | 1.285 |
| Performer | 1.293 | 1.592 | 2.239 | 2.059 | 1.588 | 2.594 | 1.558 | 1.352 |
| CosFormer | 1.191 | 1.341 | 1.987 | 1.840 | 1.330 | 2.398 | 1.443 | 1.142 |

Table 14: **KL divergence of attention weights generalizing to new data**. Hedgehog attentions trained on either CoLA (CoLA) or WikiText-103 (WT-103) data, still best match softmax attention weights computed on different GLUE tasks, despite being trained with task-specific data (measured via KL divergence; lower is better).

| Method | CoLA | MRPC | QNLI | QQP | RTE | SST-2 | STS-B |
|---|---|---|---|---|---|---|---|
| Hedgehog (CoLA) | **58.4** | **89.4** | 87.7 | 89.8 | 62.1 | **91.9** | 85.3 |
| Hedgehog (WT-103) | 47.2 | **89.4** | **89.2** | **90.4** | **62.5** | 91.4 | **86.7** |
| HH (No Train) | 50.3 | 83.3 | 85.9 | 86.5 | 55.6 | 89.5 | 79.3 |
| 1 + ELU | 26.8 | 81.9 | 78.5 | 89.1 | 55.6 | 85.9 | 41.8 |
| Performer | 24.7 | 81.4 | 75.8 | 86.5 | 55.6 | 85.1 | 39.8 |
| CosFormer | 41.1 | 82 | 82.6 | 89.3 | 54.9 | 88.4 | 76.6 |

Table 15: **Attention generalization on downstream tasks**. BERT models with Hedgehog attentions trained on either CoLA (CoLA) or WikiText-103 (WT-103) achieve best GLUE performance despite being finetuned on different GLUE tasks. This corresponds with prior observations in generalization via improved attention weight fidelity.

## C.3 LLAMA-2 SAMSUM GENERATIONS

We include sample generations from the SAMSum corpus summarization task (Gliwa et al., 2019), used to evaluate Hedgehog conversion of LLama-2 7B models in combination with low-rank adaptation (LoRA). Via the generation quality, we find that in contrast to prior conversion methods such as Transformer-to-RNN (T2R) (Kasai et al., 2021), Hedgehog makes pretrained-conversion with parameter-efficient finetuning feasible on larger models.

We report generations for four test samples of the SAMSum test set (first 3, and a longer 6th), comparing standard attention Llama-2 models, linear attention Llama-2 models achieved via Hedgehog attention distillation, and linear attention Llama-2 models following T2R, all with LoRA finetuning on a single A6000 GPU (unquantized bfloat16 weights; LoRA on all query, key, value, and output projections).

For future work, we think further showing how Hedgehog and pretrained-conversion can help apply larger models to tasks requiring longer contexts or more efficient inference, are promising avenues of exploration. Several considerations include additional explorations to improve generation quality, parameter training efficiency (*e.g.,* combining Hedgehog with quantization (Dettmers et al., 2023)), and further utilizing Hedgehog's linear attention efficiency via transfer to longer contexts (*e.g.,* combining Hedgehog with positional interpolation (Chen et al., 2023)).

```
# (Input):
 Summarize this dialog:
Hannah: Hey, do you have Betty's number?
Amanda: Lemme check
Hannah: <file_gif>
Amanda: Sorry, can't find it.
```

```
Amanda: Ask Larry
Amanda: He called her last time we were at the park together
Hannah: I don't know him well
Hannah: <file_gif>
Amanda: Don't be shy, he's very nice
Hannah: If you say so..
Hannah: I'd rather you texted him
Amanda: Just text him
Hannah: Urgh.. Alright
Hannah: Bye
Amanda: Bye bye
---
Summary:

# (Llama-2):
Hannah is looking for Betty's number. Amanda can't find it. Amanda
suggests Hannah to ask Larry.

# (Hedgehog Llama-2):
Hannah is looking for Betty's number. Amanda will text her.

# (T2R Llama-2):
along recently acquired the biggest, I'tch.SCHPW182seud is the
biggest, I't a.SCHPentua.SCHPentle't a.SCHPentua.  ?
along ch.  ?
along ch.  ?
along ch.  ?
along ch.  ?
along ch.  ?
along ch.  ?
```

Listing 5: SAMSum Llama-2 generations after LoRA finetuning.

```
# (Input):
 Summarize this dialog:
Eric: MACHINE!
Rob: That's so gr8!
Eric: I know! And shows how Americans see Russian ;)
Rob: And it's really funny!
Eric: I know! I especially like the train part!
Rob: Hahaha! No one talks to the machine like that!
Eric: Is this his only stand-up?
Rob: Idk. I'll check.
Eric: Sure.
Rob: Turns out no! There are some of his stand-ups on youtube.
Eric: Gr8! I'll watch them now!
Rob: Me too!
Eric: MACHINE!
Rob: MACHINE!
Eric: TTYL?
Rob: Sure :)
---
Summary:

# (Llama-2):
Eric and Rob are watching a stand-up by a Russian comedian.

# (Hedgehog Llama-2):
Eric and Rob are watching a stand-up comedy by a Russian comedian.

# (T2R Llama-2):
along recently acquired the biggest,
I'tch. SCHPentua. SCHPentua. SCHPentua. SCHPentua. SCHP
along ch.  ?
along ch.  ?
along ch. SCHP
along ch.  ?
along ch.  ?
along ch.  ?
along for:
along for:
al
```

Listing 6: SAMSum Llama-2 generations after LoRA finetuning.

```
# (Input):
 Summarize this dialog:
Lenny: Babe, can you help me with something?
Bob: Sure, what's up?
Lenny: Which one should I pick?
Bob: Send me photos
Lenny:  <file_photo>
Lenny:  <file_photo>
Lenny:  <file_photo>
Bob: I like the first ones best
Lenny: But I already have purple trousers. Does it make sense to have
two pairs?
Bob: I have four black pairs :D :D
Lenny: yeah, but shouldn't I pick a different color?
Bob: what matters is what you'll give you the most outfit options
Lenny: So I guess I'll buy the first or the third pair then
Bob: Pick the best quality then
Lenny: ur right, thx
Bob: no prob :)
---
Summary:

# (Llama-2):
Lenny is looking for new trousers. Bob suggests the first or the
third pair.

# (Hedgehog Llama-2):
Lenny wants to buy a new pair of trousers. Bob suggests the first two
options.

# (T2R Llama-2):
along, I't a. SCHPentua. SCHP
along ch. SCHPentua.  ?
along ch. SCHP
along ch.  ?
along ch.  ?
along ch.  ?
along for:
along ch.  ?
along ch.  ?
along for:
along ch.  ?
along for:
```

Listing 7: SAMSum Llama-2 generations after LoRA finetuning.

```
# (Input):
 Summarize this dialog:
Benjamin: Hey guys, what are we doing with the keys today?
Hilary: I've got them. Whoever wants them can meet me at lunchtime or
after
Elliot: I'm ok. We're meeting for the drinks in the evening anyway
and I guess we'll be going back to the apartment together?
Hilary: Yeah, I guess so
Daniel: I'm with Hilary atm and won't let go of her for the rest of
the day, so any option you guys choose is good for me
Benjamin: Hmm I might actually pass by at lunchtime, take the keys
and go take a nap. I'm sooo tired after yesterday
Hilary: Sounds good. We'll be having lunch with some French people
(the ones who work on the history of food in colonial Mexico - I
already see you yawning your head off)
Benjamin: YAAAAWN       Where and where are you meeting?
Hilary: So I'm meeting them at the entrance to the conference hall at
2 pm and then we'll head to this place called La Cantina. Italian
cuisine, which is quite funny, but that's what they've chosen
Benjamin: Interesting       To be honest, Hilary, I almost feel like
changing my mind. Wanting to take this nap might end up costing me to
dear
Hilary: Oh come on
Benjamin: All these terrible obstacles on mu way to bed might just
prove to much to take
Hilary: We'll try to avoid talking about their subject of research.
Oh wait, no, I'm actually meeting them because I wanted to chat about
their research lol
Elliot:
Hilary: Do join us, we're going to have fun. And then you'll take the
keys and take this most deserved of naps
Elliot: Sounds like a plan
Hilary:
Elliot: See you at 2 then xx

# (Llama-2):
Hilary has the keys. Benjamin is tired after yesterday. Hilary is
meeting some French people at 2 pm. Benjamin will join them.

# (Hedgehog Llama-2):
Hilary and Elliot are going to meet at the entrance of the conference
hall at 2 pm. Hilary and Benjamin will meet there. Hilary and
Benjamin will take the keys to the apartment. Hilary and Benjamin
will meet Elliot at 2 pm. Hilary and Benjamin will take a nap.

# (T2R Llama-2):
Most is the biggest, I's:
Most is the biggest, I's:
Most is the biggest, I's:
Most is the biggest, I's:
Most is the biggest, I's:
Most is the biggest, I's:
M
```

Listing 8: SAMSum Llama-2 generations after LoRA finetuning.

## C.4 ADDITIONAL ATTENTION WEIGHT VISUALIZATIONS

We finally include additional visualizations of the attention weights computed via softmax attention in comparison to Hedgehog and alternate linear attention feature maps. We visualize attentions computed on GLUE tasks (Sec. 5.4) from the 1st, 6th, and 12th (first, middle, last) layers of BERT models in top, middle, and bottom rows respectively, and for the 1st, 6th, and 12th heads.

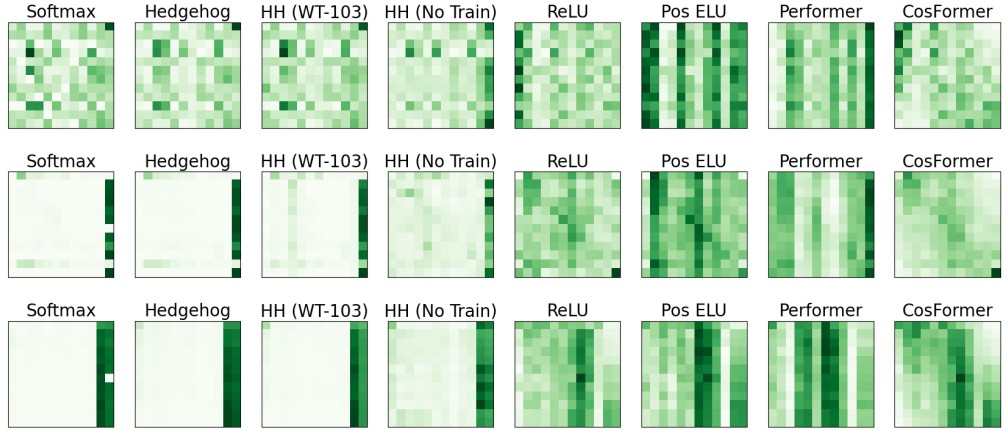

Figure 12: BERT attention visualizations for CoLA. Head 1; 1st, 6th, and 12th layers.

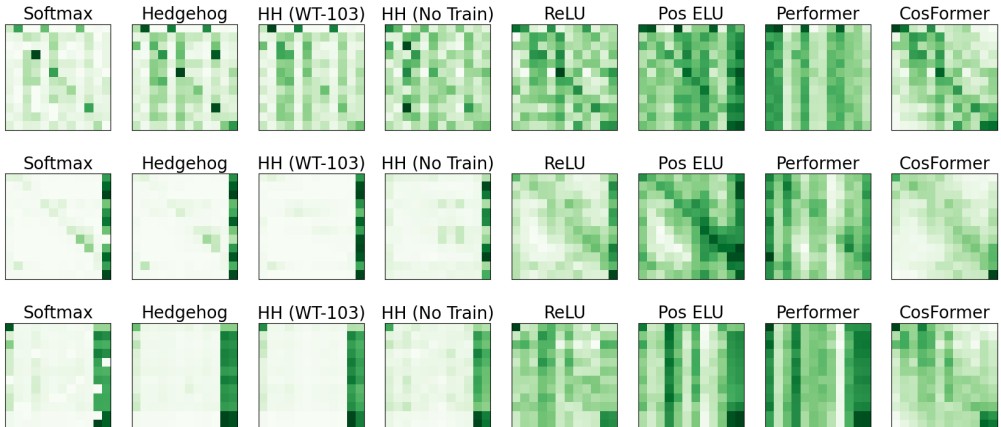

Figure 13: BERT attention visualizations for CoLA. Head 6; 1st, 6th, and 12th layers.

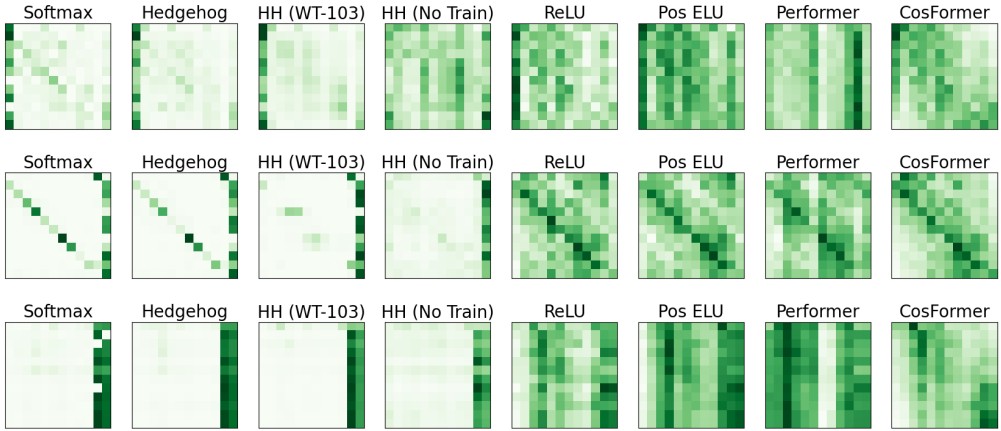

Figure 14: BERT attention visualizations for CoLA. Head 12; 1st, 6th, and 12th layers.

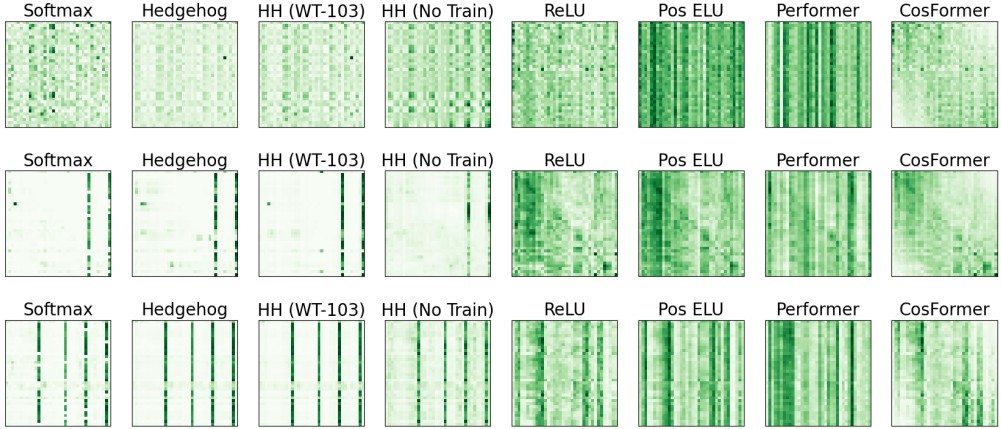

Figure 15: BERT attention visualizations for RTE. Head 0; 1st, 6th, and 12th layers.

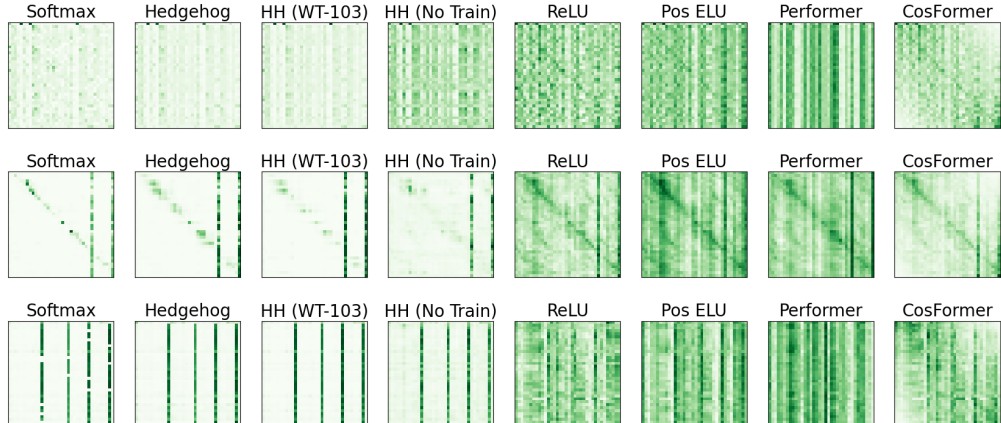

Figure 16: BERT attention visualizations for RTE. Head 6; 1st, 6th, and 12th layers.

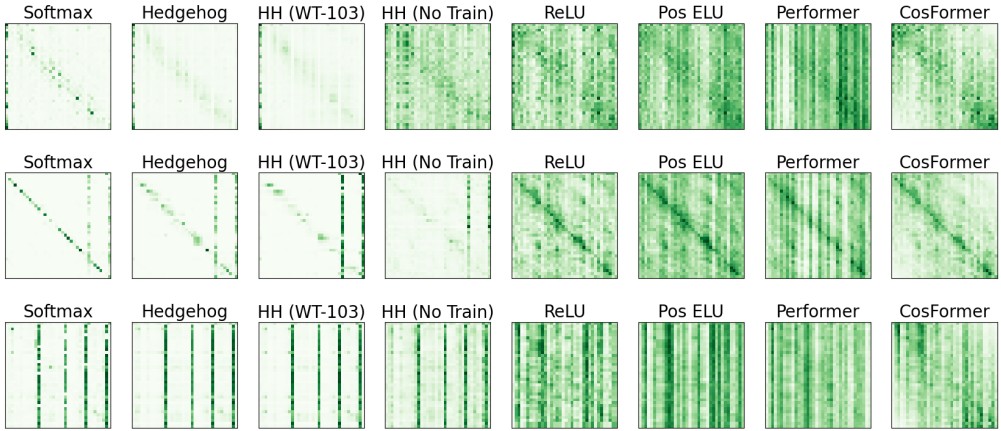

Figure 17: BERT attention visualizations for RTE. Head 12; 1st, 6th, and 12th layers.

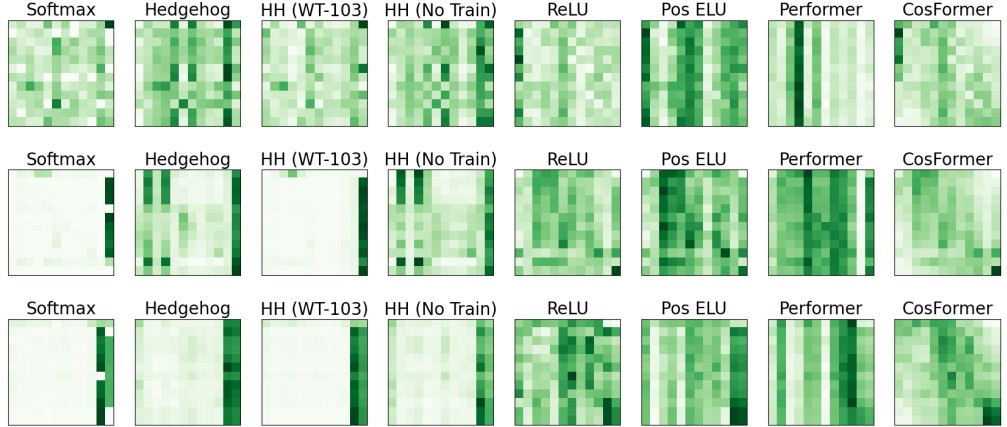

Figure 18: BERT attention visualizations for SST2. Head 1; 1st, 6th, and 12th layers.

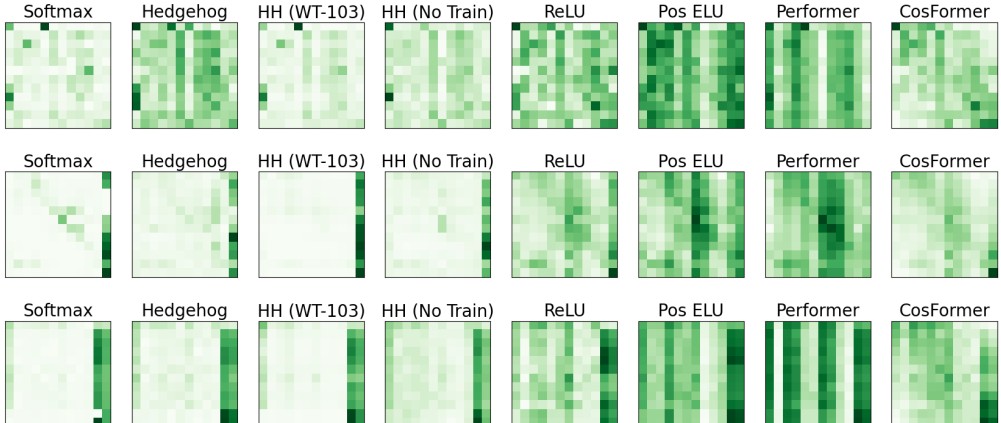

Figure 19: BERT attention visualizations for SST2. Head 6; 1st, 6th, and 12th layers.

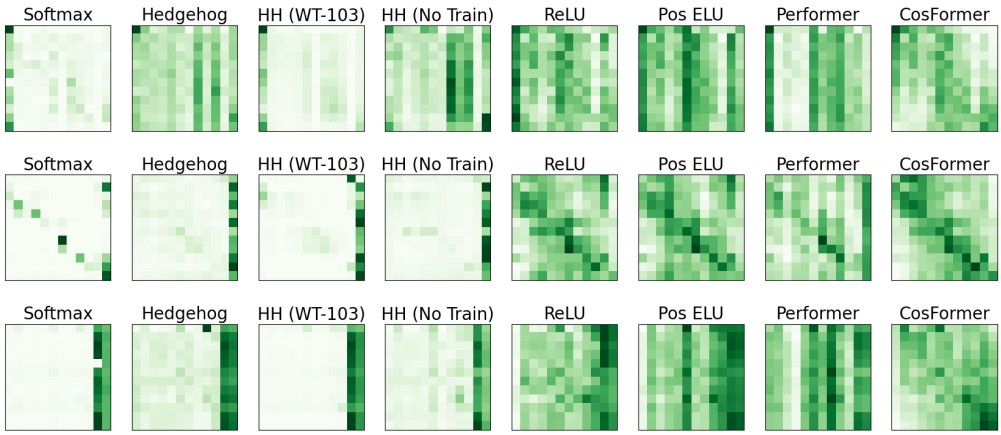

Figure 20: BERT attention visualizations for SST2. Head 12; 1st, 6th, and 12th layers.

