# OpenReview forum: "The Hedgehog & the Porcupine: Expressive Linear Attentions with Softmax Mimicry"
_ICLR.cc/2024/Conference — ICLR 2024 poster_

### Official Review · Reviewer_PsQx · 2023-11-01

**Soundness:** 3 good
**Presentation:** 2 fair
**Contribution:** 3 good
**Rating:** 6
**Confidence:** 4

**Summary:**

The authors found that prior linear attentions miss two key properties of softmax attention that are crucial for good performance: having low-entropy (spiky) weights and maintaining dot-product monotonicity. A new learnable linear attention called Hedgehog was introduced. Hedgehog retains these crucial properties while keeping linear complexity. It uses trainable MLPs to produce attention weights similar to softmax attention. The results indicate that Hedgehog can achieve over 99% of the standard Transformer's performance, outdoing previous linear attentions in various settings.

**Strengths:**

Technically sound well-written paper dealing with an important research task.

**Weaknesses:**

Table 3 is confusing since there are many more higher performing solutions for LRA: https://paperswithcode.com/sota/long-range-modeling-on-lra
Taylor expansion in transformers is well-known approach: eg. https://arxiv.org/pdf/2206.08898.pdf
While Hedgehog aims to retain the properties of softmax attention in a linear attention framework, this could introduce its own complexities in terms of implementation and optimization.

**Questions:**

Since the function mimicking softmax is learned how scalable it is to other modalities?

---

> ### Author Response · Authors · 2023-11-22
> **Response to Reviewer PsQx**
>
> Thank you for your review! We appreciate that you find our paper well-written and dealing with an important research task.
>
> We also appreciate your comments on the LRA comparisons and scalability to other modalities. We have updated the paper to address these, where we:
> 1. Clarify the scope of the LRA comparison (where we aim to compare against Transformers and attention approximations)
> 2. Clarify the framing of the Taylor approximation here vs in past work
> 2. Add additional results showing Hedgehog can apply to additional modalities (e.g., images via Vision Transformers).
> 3. Provide additional code blocks and algorithms (App A.3) to show how Hedgehog training can be implemented in simple training loops, with minimal extra complexity or optimization overhead.
>
> ---
>
> **LRA comparison**
> We clarify that although recently many non-Transformer models have shown impressive results outperforming standard (softmax) Transformers on LRA (e.g., those based on deep state-space models like S4 [1]), because our work focuses on how to improve linear attentions and fits in the Transformer framework, we focus the comparison against other attention-based methods.
>
> We have updated the framing in Sec 5.2. to clarify this.
>
> [1] Efficiently Modeling Long Sequences with Structured State Spaces (Gu et al., 2021).
>
> ---
>
> **Taylor approximation**
> Thanks as well for the comments on Taylor series approximations being well-known. We updated Sec. 4.1 to reference the original citations. We would also like to clarify that although prior works explore Taylor series approximations for the exponential, they do so under different motivations, (e.g., as linked, as a replacement for the softmax used in the final layer of a classification network).
> * Here we use the Taylor Series approximation to empirically study its properties for a good linear attention mechanism. We further point out the practical inefficiencies with the Taylor approximation with respect to prior linear attentions, and use these limitations to motivate the development of our Hedgehog method.
> ---
>
> **Scaling to other modalities**
> Thanks for this question. We ran additional experiments for Hedgehog applied to vision, and found Hedgehog can recover 99% of pretrained Vision Transformer top-1 accuracy when doing finetuned conversion for ImageNet-1K.
>
> We use the readily available ViT-base model (vit-base-patch16-224), and find that while the original softmax attention version gets 80.3% top-1 accuracy, Hedgehog obtains 79.5% top-1 accuracy. We added additional experimental details and comparisons in the updated draft (Section 5.4, Appendix B.4).
>
> ---
>
> **Hedgehog implementation complexity**
> We updated the appendix (App. A.3) to show how Hedgehog training can be performed in a simple training loop, where we can jointly optimize all Hedgehog MLPs with the same optimizer in an end-to-end fashion. We make use of popular pretrained Transformer APIs such as HuggingFace transformers to output the attentions at each layer.
>
> This enables training without much more complexity than a two-step process where we (1) freeze all original model parameters, insert the Hedgehog feature maps, and train the Hedgehogs end-to-end with the attention distillation loss, before (2) unfreezing the original model parameters and training the entire model end-to-end on the task-loss.

---

### Official Review · Reviewer_y6Q1 · 2023-11-02

**Soundness:** 3 good
**Presentation:** 4 excellent
**Contribution:** 3 good
**Rating:** 8
**Confidence:** 4

**Summary:**

This paper proposes *Hedgehog*, a new linear attention mechanism for Transformers.

The authors start by noting that previous linear attention mechanisms still lag considerable behind full quadratic attention. The authors then investigate why this is the case, by studing the properties of the attention maps produced by full attention and linear approximations. They find that two properties in full attention that previous linear attention methods seem to lack:

- low-entropy "spiky" attention weights
- monotonicity over query-key dot products

The authors then attempt to measure the importance of these two properties by inducing these properties in some test tasks, finding that these properties seem to correlate with performance of the trained model.

Finally, the authors propose Hedgehog: the core idea being using linear map from the query/key space into a lower-dimensional feature space, but learning this linear map to *explicitly* approximate the final attention maps produced by dense attention.

Experimental results are quite convincing: Hedgehog outperforms most other linear attention methods on multiple tasks (including LM perplexity and popular downstream tasks), both when training models from scratch and when “linearizing” models with dense attention.

**Strengths:**

- The paper provides a quite consistent story: they identified a problem (current linear-attention LLMs/transformers don’t work that well), studied potential reasons for their failure mode and verified their importance for the performance of models (the two properties of attention) and proposed and experimentally verify a fix to these problems that leads to linear-attention models that get closer to dense attention models)
- The paper is well-written, and the problem studied very relevant: the quadratic complexity is one of the main bottlenecks preventing the scaling of transformers to longer contexts.
- The identified properties of dense attention (and the subsequent analysis of their relation to downstream performance) are super interesting (and could be a contribution almost independently of the proposed method)
- The proposed Hedgehog method seems to be effective for the model-size studied in this work, with Hedgehog leading to much less performance degradation than other linear-attention approaches (when compared to dense-attention model)

**Weaknesses:**

- My main criticism of the study is that they only studied relatively small transformer models:  they only studied 125M decoder only models. Previous work [1] has shown that linear attention approach are less amenable to scaling than dense attention (meaning as models get larger, the gap between dense and linear attention increases) and so it’s unclear if this approach would still lead to such small gaps with dense attentions for 1B+ models (which IMO, is the main use case for this technique). While I already really liked the paper, I think this paper could be much stronger impact on the community if results were validated on larger LLMs.

[1] https://arxiv.org/pdf/2207.10551.pdf

**Questions:**

- In the train-from-scratch setting, how is the Hedgehog attention initialize?. From my understanding, Hedgehog requires an already-trained dense attention model to finetune the feature map. Does this imply that in this scenario we still need to train a dense attention model anyway?

---

> ### Author Response · Authors · 2023-11-22
> **Response to Reviewer y6Q1**
>
> Thank you very much for your review! We are glad that you appreciated our work, and are likewise excited about scaling Hedgehog up to larger models. We address your questions on this scaling and clarification of Hedgehog initialization when training-from-scratch below.
>
> **Hedgehog scaling to Llama-2 7B**
> Inspired by your comments, we added initial experiments showing that Hedgehog enables conversion of Llama-2 7B models into linear attention versions, where subsequent parameter-efficient finetuning results can significantly improve performance over the base model on a downstream summarization task. Furthermore, we found prior attempts to swap the attentions for linear attentions did not seem to work.
>
> We describe these findings below, but please find additional results (Sec. 5.4, Table 11; Appendix C.3)  and experimental details (Appendix B.5) in our updated draft.
>
> ***Llama-2 linearization setup***
> To do the Llama-2 linearization, we first train Hedgehog MLPs to match the attentions layer-by-layer and head-by-head with frozen Llama-2 weights on downstream task data (e.g., the SAMSum summarization corpus [1]). Doing so amounts to training 0.5% of the original model parameters. Following this attention distillation, we replace the original Llama attentions with the Hedgehog attentions (keeping the original Llama weights for query, key, value, and output projections, but computing the QKV interactions via linear attention after applying the Hedgehog feature maps).
>
> We then apply low-rank adaptation (LoRA) [2] to the query, key, value, and output projections, and finetune the entire model on the downstream task (summarization).
>
> ***Llama-2 linearization results***
> Table 1 shows our results, where we report ROUGE metrics (unigram, bigram, longest common subsequence) for test set generations. As baselines, we compare with zero-shot Llama-2 model and Llama-2 with LoRA. We also compare with linearizing Llama-2 using the prior linear attention conversion method, Transformer-to-RNN (T2R), followed by LoRA finetuning.
>
> |                     |  ROUGE-1  |  ROUGE-2  |  ROUGE-L  |
> |---------------------|:----:|:----:|:----:|
> | Softmax (Zero-shot) | 19.3 |  6.8 | 14.9 |
> | Softmax (LoRA)      | 51.1 | 27.6 | 43.5 |
> | T2R (LoRA)          |  2.8 |  0.0 |  2.6 |
> | Hedgehog (LoRA)     | 47.4 | 23.4 | 39.1 |
>
> While there exist a smaller quality gap with the full softmax-attention Llama-2 with finetuning, we find Hedgehog enables linear attention finetuning that significantly improves the base zero-shot performance (e.g., 14.9 to 39.1 ROUGE-L). Meanwhile, we struggled to get prior linearization techniques to work (2.6 ROUGE-L with T2R).
>
> For more granular evaluation of summarization quality, we encourage viewing the sample generations among the LoRA-finetuned models in Appendix C.3. Here we find that the Hedgehog-Llama-2 models are able to produce coherent summaries similar to standard Llama-2, while the T2R-Llama-2 models struggle to generate coherent text.
>
>
> [1] SAMSum Corpus: A Human-annotated Dialogue Dataset for Abstractive Summarization (Gliwa et al., 2019)
>
> [2] LoRA: Low-Rank Adaptation of Large Language Models (Hu et al., 2021)
>
> ---
>
> **Clarification on Hedgehog attention initialization when training-from-scratch**
> We clarify that for training-from-scratch, we actually do not do the first step of attention-weight distillation. We instead initialize the Hedgehog MLPs as identity matrices (keeping the exponential feature map form discussed in Eq. 3), and train the entire model end-to-end just with the task-specific loss. Thus in this case we do not need to train a dense attention model first.
>
> We updated the draft to clarify this in Section 5.2, and provide additional implementation details with pseudocode in Appendix A.2 and A.3.

---

### Official Review · Reviewer_ZtxG · 2023-11-02

**Soundness:** 3 good
**Presentation:** 2 fair
**Contribution:** 3 good
**Rating:** 5
**Confidence:** 4

**Summary:**

This paper address the quadratic scaling of attention mechanism in transformers. There are 2 main contributions:

1. Defining characteristics that make an attention mechanism work well: (a) low-entropy "spikyness" and (b) dot-product monotonicity

2. Two new attention mechanisms to evaluate their claims. Taylor and Hedgehog

 They evaluate attention mechanisms in several scenarios (1) Training-from-scratch, (2) Finetuned-conversion, and (3) Pretrained-conversion.

**Strengths:**

The exploration into "spikyness" and monotonicity is interesting and novel to me.

The regime for evaluating their attention mechanism is thorough and covers a wide variety of situations.

The ablations are well done and the experimental results on benchmarks and efficiency are strong.

**Weaknesses:**

I found the presentation a little confusing. In some figures and tables only Hedgehog is shown. In others only Taylor is shown. Should both be evaluated?

Is the GPT-2 column in Table 6 unfair? It seems they should finetune with the full attention mechanism?

**Questions:**

There doesn't seem to be much downside to this method except for training from scratch? Should everyone be finetuning and converting their pretrained models to Hedgehog?

Was there any investigation into the learned weights of Hedgehog to gain intuition on how it's approximating softmax?

How does Hedgehog's softmax approximation do with length generalization (sequence length longer than training data) or data that is out of distribution?

---

> ### Author Response · Authors · 2023-11-22
> **Response to Reviewer ZtxG (1/2)**
>
> Thank you for your review! We appreciate finding our work interesting and novel, with thorough and strong benchmarks.
>
> We also appreciate the feedback on presentation on comparisons between the Taylor approximation for exponentials and Hedgehog, and comparison to GPT-2. We first address these below, and have updated the draft to reflect these changes.
>
> In the next reply thread, please find our answers to your questions on Hedgehog generalization and understanding of the learned weights. We appreciate the insightful questions, and have updated both the main paper and appendix to include additional analysis inspired by your comments.
>
> **Comparison between Hedgehog and Taylor Series exponential approximation**
> We clarify that we study the Taylor Series approximation as *motivation* for our Hedgehog method. We primarily aim to recover the desirable expressive traits of the Taylor approximation, while overcoming its efficiency limitations (despite being computable as a linear attention, the Taylor approx. introduces extra inefficiencies via larger feature dimension (c.f. “Caveats” in Section 4.1).
>
> As such, for our main results we only evaluate Hedgehog.
>
> From your feedback, we updated our motivating analysis in Section 3 to show that Hedgehog performs comparably on all the Taylor series results.
> * In particular, we update Fig. 2 and 3 to show Hedgehog indeed recovers the spiky and monotonic properties, and update Table 3 to show that Hedgehog also solves the associative recall task.
>
> **Comparison to GPT-2 column in Table 6**
> Thanks for pointing this out. We updated Table 6 (now Table 10) to also include a finetuned version of GPT-2 with the full attention mechanism (reproduced below).
>
> While the fully finetuned GPT-2 gets lowest PPL at 15.8, this incurs relatively more expensive quadratic attention during finetuning. Meanwhile, Hedgehog still significantly closes the gap among subquadratic models, via finetuning with only linear attention.
>
> Originally, we intended the evaluation in Table 6 to test if Hedgehog enables significant downstream transfer, i.e., we can successfully finetune the linearized GPT-2 on new tasks it is not already good at. Hence we included the zero-shot GPT-2 as a baseline reference, where the large improvement in perplexity (from 28.0 to 16.7) suggests Hedgehog not only enables recovering the base capabilities (As explored in Sec 5.3 with the BERT models), but also further supports finetuning on new tasks.
>
> We included comparisons to other subquadratic architectures (H3, Hyena) to show the effectiveness of this pretrained conversion; i.e., instead of training a new subquadratic model from scratch, we can start with existing powerful but expensive-to-compute quadratic models, and get a “best-of-both-worlds” situation with Hedgehog by leveraging the pretrained weights but via more efficient linear attention.

---

> ### Author Response · Authors · 2023-11-22
> **Response to Reviewer ZtxG (2/2)**
>
> **Investigation into the learned weights of Hedgehog for intuition**
> We explored this under two angles and added our findings in the updated draft.
>
> First, we tried to better understand if the learned weights are actually important, or if just the exponential feature map suffices. We expanded Sec. 4.3 (now Sec. 5.1) to visualize the attention weights of Hedgehog without training (Fig. 8) in comparison to with training (Fig. 7) and found that learning the MLP weights (vs just replacing the MLPs with the identity matrix) lead to significantly better matching of softmax attentions. Please see additional visualizations in Appendix C.3 and C.4 for additional results.
>
> We also tried visualizing the actual MLP weights of trained Hedgehog layers, but were not able to find any discernible patterns or insights as of yet. We definitely think further analysis into how these weights “shape” the softmax over time could be interesting to explore for additional understanding.
>
> **How does Hedgehog's softmax approximation do with length and data generalization?**
> Thanks for these questions! We explored these further in our updated section on benchmarking Hedgehog (now Sec. 5.1) and added additional studies in the appendix (C.2.1 - C.2.3) to dig deeper. We describe some findings below, but please find our full investigations in the updated draft.
>
> ***New data generalization***
> First, on new data, we find Hedgehog feature maps learned on one dataset can continue to approximate softmax attentions for another. We trained Hedgehog MLPs for BERT models on the CoLA GLUE task, and evaluated how well the resulting attention weights would match in comparison to softmax attention weights on different GLUE tasks (using KL divergence to measure). We also compare with the attentions using alternative linear attention feature maps from prior work (1 + ELU, Performer, CosFormer).
>
> Table 2 below shows that despite being trained to match softmax attention weights over CoLA data, Hedgehog feature maps continue to better match the softmax attentions than alternative linear attentions (reporting KL divergence, lower is better; please see Fig. 9, 10, 11 for attention weight visualizations, Table 14 for results in updated draft, Table 4 for abridged in main paper). We do note a drop in fidelity (higher KL divergence) and think investigating and overcoming the source of this would be exciting future work.
>
> | Method          |  CoLA |  MNLI |  MRPC |  QNLI |  QQP  |  RTE  | SST-2 | STS-B |
> |-----------------|:-----:|:-----:|:-----:|:-----:|:-----:|:-----:|:-----:|:-----:|
> | Hedgehog (CoLA) | 0.173 | 0.340 | 0.652 | 0.673 | 0.374 | 0.932 | 0.211 | 0.275 |
> | 1 + ELU         | 1.231 | 1.500 | 2.150 | 1.947 | 1.509 | 2.491 | 1.505 | 1.285 |
> | Performer       | 1.293 | 1.592 | 2.239 | 2.059 | 1.588 | 2.594 | 1.558 | 1.352 |
> | CosFormer       | 1.191 | 1.341 | 1.987 | 1.840 | 1.330 | 2.398 | 1.443 | 1.142 |
>
> ***Longer length generalization***
> Second, on new lengths, we find that the lower KL divergence achieved via Hedgehog also remains consistently low when transferring to longer samples. For this, we first combine individual CoLA samples into longer sequences up to 4096 tokens long. We then evaluate how Hedgehog MLPs trained on individual CoLA samples generalize to matching the softmax attentions when computed over these longer sequences in Table 3 below (Table 5 in updated draft).
>
> | Sequence Length |  256  |  1024 | 2048 |  4096 |
> |-----------------|:-----:|:-----:|:----:|:-----:|
> | KL Divergence   | 0.182 | 0.187 | 0.19 | 0.181 |
>
> We think these questions on generalization are particularly exciting, perhaps motivating an interesting line of work for “pretrained” linear attention feature maps. Ideally, we could train feature maps once, e.g., distilling the attention weights from some base Transformer model over some pretraining corpus (perhaps similar to pretraining large LLMs today), and seeing if the learned feature maps could then be a “universal” drop-in replacement for softmax attentions (applicable to new data, but also perhaps new models).

---

### Author Response · Authors · 2023-11-22
**General Response + Revision Summary**

Many thanks to the reviewers for their helpful comments and feedback.
* We appreciated that reviewers consistently found our work interesting and novel (ZtxG, y6Q1), tackling a relevant and important research problem of improving Transformer efficiency via linear attention (y6Q1, PsQx).
* We also thank reviewers for noting our thorough, consistent, and well-written investigation, and finding our method (Hedgehog) effective on a wide variety of evaluation settings (ZtxG, y6Q1, PsQx).

Thanks to reviewers comments, we updated the draft to include additional results, experiments, and analysis on Hedgehog’s performance (expanding to new data, modalities, models), along with general tidying of the presentation (changes in blue).

These include:

#### **1. Extended analysis of Hedgehog generalization to new data and sequence lengths**

* Thanks to Reviewer ZtxG, we updated Section 4.3 (now Section 5.1) to include new results showing Hedgehog’s learned feature maps can transfer to new data. We extend this analysis in Appendix C.2 to show this corresponds with improved downstream task performance.

* We also show that Hedgehog’s attention weights continue to match softmax attention over inputs with longer sequence lengths.

#### **2. Further support for new models and modalities**

* We include additional results showing that Hedgehog can be applied to additional models and modalities via Llama-2 7B and ViT-base (patch size 16) linear attention conversion.

  * Thanks to Reviewer PsQx’s comment on different modalities, we show Hedgehog supports linear attention conversion of Vision Transformers, recovering 99% of top-1 ImageNet-1k accuracy for the popular pretrained ViT-base model (Section 5.3)

  * Thanks to Reviewer y6Q1’s comment on scaling up, we show Hedgehog also works to convert Llama-2 7B models for downstream finetuning. In particular, we can combine Hedgehog with parameter-efficient finetuning (e.g., LoRA), and show that we can finetune the “linearized” Llamas for downstream summarization tasks (Section 5.4).

#### **3. General tidying of presentation**

We appreciated reviewers’ helpful comments and feedback, and made best efforts to incorporate and address all in the latest revision.

* We revised the presentation in our motivating analysis (Section 3) to make clear the relation between Hedgehog (our method) and alternatives that motivate Hedgehog’s development (such as the Taylor approximation for exponentials, which we clarify is noted in prior work)

* Relatedly, we clarify the evaluation settings and scope of evaluation for individual experiments

* We extend the Appendix to include additional results, analysis, and experiments to complement the main findings presented in the main paper.


Please see our responses to individual reviewers in the threads below for details. We are happy to follow-up and discuss any remaining questions or comments.

---

### Meta-Review · Area_Chair_ErVW · 2023-12-14

**Metareview:**

This paper provides new frameworks for linear attention to avoid the quadratic bottleneck of transformers. It characterizes two properties of softmax attention, low-entropy (spikiness) and monotonicity. They show prior approaches to linear attention to do not possess these properties and propose to learn the linear approximation. Empirical performance is stronger than previous approaches. There is some disagreement about this paper, with two reviewers feeling positive and one feeling negative about it. I believe the author rebuttal and revised version addressed the concerns of the latter, while also applying their proposed technique to larger models (LLaMA2, 7B). I am leaning towards acceptance.

**Justification For Why Not Higher Score:**

It is not a tremendously original idea.

**Justification For Why Not Lower Score:**

While most previous linear attention models didn't work very well, the proposed method seems consistently better.

---

### Decision · Program_Chairs · 2024-01-16

Accept (poster)